# Integrated analysis of Xist upregulation and X-chromosome inactivation with single-cell and single-allele resolution

Guido Pacini[1], Ilona Dunkel[1], Norbert Mages[2], Verena Mutzel [1], Bernd Timmermann[2],
Annalisa Marsico [3,4 ✉] & Edda G. Schulz [1,4 ✉]

To ensure dosage compensation between the sexes, one randomly chosen X chromosome is silenced in each female cell in the process of X-chromosome inactivation (XCI). XCI is initiated during early development through upregulation of the long non-coding RNA Xist, which mediates chromosome-wide gene silencing. Cell differentiation, Xist upregulation and gene silencing are thought to be coupled at multiple levels to ensure inactivation of exactly one out of two X chromosomes. Here we perform an integrated analysis of all three processes through allele-specific single-cell RNA-sequencing. Specifically, we assess the onset of random XCI in differentiating mouse embryonic stem cells, and develop dedicated analysis approaches. By exploiting the inter-cellular heterogeneity of XCI onset, we identify putative Xist regulators. Moreover, we show that transient Xist upregulation from both X chromosomes results in biallelic gene silencing right before transitioning to the monoallelic state, confirming a prediction of the stochastic model of XCI. Finally, we show that genetic variation modulates the XCI process at multiple levels, providing a potential explanation for the long-known X-controlling element (Xce) effect, which leads to preferential inactivation of a specific X chromosome in inter-strain crosses. We thus draw a detailed picture of the different levels of regulation that govern the initiation of XCI. The experimental and computational strategies we have developed here will allow us to profile random XCI in more physiological contexts, including primary human cells in vivo.

[1] Otto Warburg Laboratories, Max Planck Institute for Molecular Genetics, Berlin, Germany. [2] Sequencing core facility, Max Planck Institute for Molecular Genetics, Berlin, Germany. [3] Institute for Computational Biology, Helmholtz Center, München, Germany. [4] These authors contributed equally: Annalisa Marsico, Edda G. Schulz. ✉email: annalisa.marsico@helmholtz-muenchen.de; edda.schulz@molgen.mpg.de

Female therian mammals carry two X chromosomes, while males have an X and a Y. Gene dosage differences between the sexes for X-linked genes are mostly compensated through X-chromosome inactivation (XCI). In this process, each female cell will silence one randomly chosen X chromosome in a cell-autonomous fashion[1]. Since XCI induces differential gene activity at two genetically identical chromosomes in the same nucleus, it is an important model for epigenetic gene regulation. The regulatory principles that allow the two X chromosomes to assume opposing activity states are only starting to be elucidated[2].

A subset of X-linked genes are incompletely silenced on the inactive X chromosome and can thus escape XCI[3]. These escape genes are thought to contribute to phenotypic differences between the sexes, including susceptibility to various pathologies, such as autoimmune diseases[4]. Moreover, inter-individual variability, for example with respect to the severity of X-linked diseases, is observed in cases where the choice of the inactive X chromosome is skewed through genetic polymorphisms[5]. Whether and to what extent differences in silencing efficiency and escape propensity driven by genetic variation might also contribute to phenotypic variability in humans remains unknown.

XCI is established during early embryonic development in a complex multi-step process. It is initiated by upregulation of the long non-coding RNA Xist, the master regulator of XCI[1]. Xist will coat the X chromosome, from which it is expressed, and will initiate chromosome-wide gene silencing by recruiting a series of silencing complexes, ultimately leading to heterochromatinization of the entire chromosome[6]. To ensure inactivation of a single X chromosome, Xist expression must be restricted to exactly one out of two alleles, in a monoallelic (MA) and female-specific fashion. While the majority of cells directly upregulate Xist monoallelically, we and others have recently shown that a subset of cells transiently express Xist from both chromosomes in a BA (BA) manner, which is subsequently converted to a MA state[7,8]. The current model is thus that Xist is initially upregulated independently on each chromosome in a random fashion and that establishment of a MA state is then ensured through negative feedback regulation[2]. This feedback is thought to be mediated by silencing of an essential X-linked Xist activator[9,10]. It remains unknown, however, to what extent transient BA Xist upregulation indeed induces gene silencing, which is a prerequisite for the proposed negative feedback to work. A prediction from this "stochastic model of XCI" is that accelerated upregulation of Xist from one allele for instance caused by genetic variation will lead to preferential inactivation of that chromosome.

Random XCI is initiated during early embryonic development around the time of implantation into the uterus, when cells exit the pluripotent state. A series of factors have been implicated in triggering developmental Xist upregulation, such as the Xist activator Rnf12/Rlim, Xist's repressive antisense transcript Tsix and a series of pluripotency factors, such as Nanog, Oct4/Pou5f1, Sox2, Klf4, and Rex1/Zfp42[11-16]. Pluripotency factors are indeed downregulated concomitantly with Xist upregulation and loss-of-function perturbations have been shown to increase Xist expression for several of them[12,13,17]. It remains unknown, however, which factors trigger Xist upregulation in the endogenous context and whether this is mediated through modulating the activity of other Xist regulators such as Rnf12 or Tsix.

The different processes governing XCI have been studied extensively and we slowly see a picture emerging of how inactivation of one randomly chosen X chromosome might be achieved. However, many studies have been performed in engineered systems to be able to investigate a specific step in isolation[18-20]. As a consequence a series of questions remain unanswered, which must be investigated in the endogenous context of random XCI with sufficient cellular, allelic and temporal resolution. Those questions include (1) to what extent gene silencing occurs upon BA Xist upregulation, (2) how differences in speed of Xist upregulation will affect the choice of the inactive X, and (3) which factors actually trigger Xist expression.

In this work, we therefore perform an integrated analysis of Xist upregulation and gene silencing with single-cell resolution, in a context where cells make a random choice between their two X chromosomes. To this end we profile the onset of XCI using allele-resolved single-cell RNA-sequencing (scRNA-seq). We use differentiating mouse embryonic stem cells (mESC), the classic tissue culture model of XCI. Since the cell line used has been derived from a hybrid cross between distantly related mouse strains, we can distinguish the two X chromosomes through single nucleotide polymorphisms (SNPs) and in addition investigate how genetic variation affects the different steps that govern XCI. Building on this high-quality dataset, we develop a computational framework to study the heterogeneity of Xist expression, XCI dynamics and differentiation over time at the single-cell level. Specifically, we analyze the full transcriptome heterogeneity throughout the estimated pseudotime and use RNA velocity to annotate lineage trajectories corresponding to the choice of the inactive X chromosome. By exploiting the variability in gene expression across cells we recover known regulators of Xist, but also identify potential novel Xist repressors and activators. We find that XCI is initiated on both chromosomes in a subset of cells through BA Xist upregulation, but is rapidly reversed to a MA state, when silencing is initiated. By computing Xist expression and gene-silencing dynamics in an allele-specific (AS) manner, we discover that genetic variation modulates XCI at multiple levels, including expression frequency of Xist as well as chromosome-wide silencing efficiency and escape propensity of individual genes. Finally, we validate these findings through an orthogonal experimental approach. Our study thus provides a detailed view on how random X inactivation is first established. The approaches we develop will also be useful to profile endogenous XCI in other contexts, such as primary human tissues.

## Results

**X inactivation and X upregulation in differentiating mESCs.** At the onset of XCI, the inactive X chromosome is chosen in a random cell-autonomous process, resulting in a mixture of cells that have silenced the paternal or the maternal X chromosome, respectively. To investigate the onset of random XCI with single-cell resolution, we profiled female mESCs at multiple time points during differentiation by 2i/Lif withdrawal using scRNA-seq (Fig. 1a). To allow AS transcript quantification, we used a F1 hybrid mESC line (TX1072), which has been derived by crossing two distantly related mouse strains, *Mus musculus domesticus* (C57BL6/J) and *Mus musculus castaneus* (Cast/EiJ), herein referred to as B6 and Cast, respectively[21]. At multiple time points during differentiation (0–4 days) we captured in total 1945 individual cells on a C1 microfluidics system (Fluidigm). Single-cell transcriptomes were profiled using the C1-HT protocol, which performs 3′-end counting using unique molecular identifiers (UMI). In contrast to previous studies, which had used the unstranded full-length Smart-seq2 protocol[22,23], 3′-end counting allows us to distinguish Xist from its antisense transcript Tsix through strand-specificity. Around 0.4 Mio reads were sequenced per cell, resulting in a median number of 0.12 Mio unique UMI counts, which represent the detected mRNA molecules and covered a median number of 6090 genes (248 X-linked) per cell (Supplementary Fig. 1a, b). After removing low-quality cells and cells that had lost one X chromosome, between 257 and 341 cells were retained per time point for further analysis (Supplementary Fig. 1b, c, Supplementary Data 1).

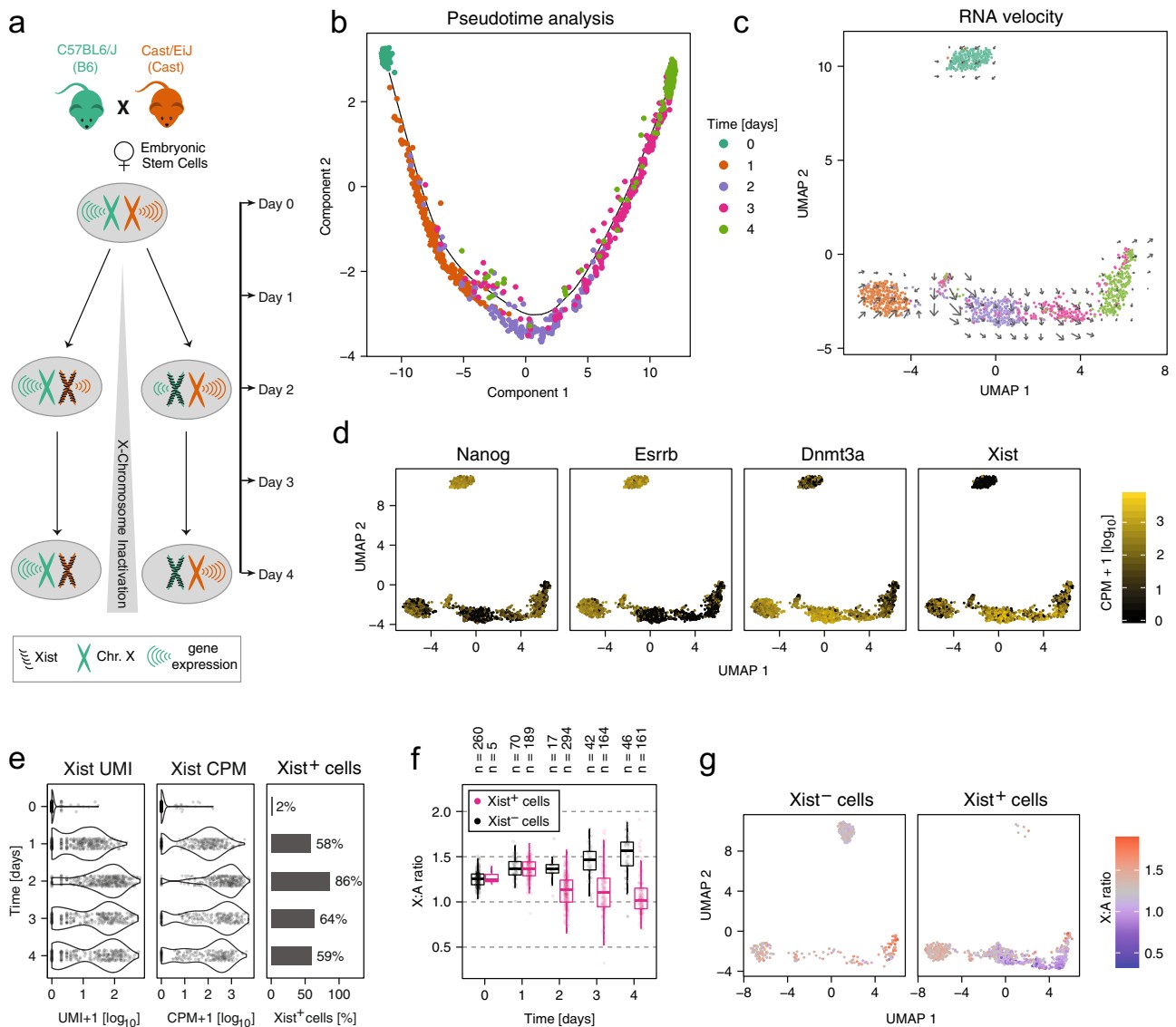

**Fig. 1 Profiling the onset of random XCI by scRNA-seq. a** Schematic representation of the experimental setup. A female mESC line derived from the cross between a B6 (green) and a Cast (orange) mouse was differentiated for 4 days by 2i/Lif withdrawal and up to 400 single-cell transcriptomes were collected per time point. During the time course, cells initiate random XCI by monoallelic upregulation of Xist (black) from one randomly chosen allele, which will induce chromosome-wide gene silencing. **b**, **c** Pseudotime analysis (**b**) and UMAP embedding (**c**) based on the 500 most variable genes, with individual cells colored by measurement time. The black line in **b** represents the principal graph describing the pseudotime trajectory of the projected cells as computed by the Monocole2 DDRTree method. Arrows in **c** indicate the predicted transcriptome change estimated through RNA velocity analysis. **d** UMAP embedding as in **c** with cells colored according to marker gene expression. **e** Distribution of Xist expression across cells, either shown as the number of UMI counts (=number of molecules, left), the normalized CPM value (middle) or the percentage of Xist-positive (>5 Xist UMI counts) cells (right). **f** Box plot of the X-to-autosome expression ratio in Xist-positive (pink) and Xist-negative cells (black). The central mark indicates the median, and the bottom and top edges of the box indicate the first and third quartiles, respectively. The top and bottom whiskers extend the boxes to a maximum of 1.5 times the interquartile range. Dots represent individual cells, the number of cells in each group is given on top. **g** UMAP embedding as in (**c**) with Xist-positive and Xist-negative cells colored according to X-to-autosome expression ratio. Color scale was centered (gray) on the median X-to-autosome expression ratio observed at day 0. Source data are provided as a Source Data file.

In a first step, we analyzed how the cells' transcriptomes changed over the time course measured. Pseudotime analysis with Monocle[24] and dimensionality reduction using UMAP[25] revealed that undifferentiated mESCs clustered distantly from their differentiating derivatives, suggesting that the change of culture condition induced a major change in their transcriptomes (Fig. 1b, c). Pseudotime analysis resolved the sampling time of most cells during differentiation (Fig. 1b). To assess the lineage trajectory of each cell we performed an RNA velocity analysis, which estimates transcriptional activity from reads aligning to

introns (unspliced) and uses that information to predict how mature mRNA expression will change in the future (Fig. 1c, arrows, Supplementary Fig. 1a)[26]. The results suggested that cells moved along a single differentiation trajectory. This was confirmed by an analysis of marker genes revealing that naive pluripotency factors such as Nanog and Esrrb were down-regulated, while Dnmt3a, a marker of primed pluripotency was upregulated (Fig. 1d).

Analysis of Xist expression in the UMAP projection showed the expected upregulation during differentiation, but also revealed

marked heterogeneity with a subset of cells expressing no or low levels of Xist (Fig. 1d, right). Xist was detected with >5 UMI counts (Xist-positive cells) in only 2% of undifferentiated mESCs, but in 58–86% of cells during differentiation, with the expression level varying strongly between cells and between time points from around 10 molecules at day 1 to >100 molecules (=UMI counts) at later stages (Fig. 1e). Xist expression appeared to be maximal at day 2 and decreased at later time points. We estimated the detection rate in our data set to lie around 30%, given that the number of mRNAs present in an ES cell has been estimated to be around ~400,000 molecules[27], ~120,000 of which we detected per cell (Supplementary Fig. 1b). The actual mean copy number of the Xist RNA in Xist-expressing cells would thus increase from 79 at day 1 to 243–314 at the later time points, which is in good agreement with a previous estimate of ~300 molecules[28].

When analyzing the read distribution along genes, we noticed an unusual read pattern for Xist. Instead of the expected 3′ bias (see examples in Supplementary Fig. 2a–c), we observed a strong peak in Xist's first exon, but did not detect a robust signal at its 3′-end (Supplementary Fig. 2d, e). A similar read pattern was also found in a previously published data set, which used CEL-seq2, a different 3′-end scRNA-seq method, to profile mouse fibroblasts (Supplementary Fig. 2f)[29]. Inspection of the sequence upstream of the peak revealed a genomically encoded poly-adenine (polyA) stretch within the Xist RNA, which likely served as a template to prime reverse transcription at this position (Supplementary Fig. 2d). The fact that read density in Xist's first exon showed the expected temporal profile (upregulation over time, Fig. 1e) suggested that Xist was nevertheless correctly quantified in our data set, even though its polyA tail appeared to be inaccessible to the reverse transcription reaction. To assess whether we could observe Xist-induced gene silencing, we estimated expression from the X chromosome relative to autosomal genes (X:A ratio, Fig. 1f). We classified cells as Xist-positive, if >5 UMI counts were assigned to the Xist gene and as Xist-negative, if Xist was not detected in the cell. Starting from day 2 of differentiation we observed a clear downregulation of X-linked genes in Xist-positive cells, from a median X:A ratio of 1.25 across all cells before differentiation to 1.03 at day 4 in Xist-positive cells (Fig. 1f, g, Supplementary Fig. 3a, $p < 0.001$, Mann–Whitney $U$ two-sided test). At the same time, the X:A ratio increased to 1.58 in Xist-negative cells, which presumably failed to initiate XCI ($p < 0.001$, Mann–Whitney $U$ two-sided test). Also the median expression across all X-linked genes relative to autosomal expression increased over time, suggesting a chromosome-wide effect (Supplementary Fig. 3b). Xist-negative cells upregulated differentiation markers, such as Dnmt3a, and downregulated pluripotency factors, such as Nanog and Esrrb (Supplementary Fig. 3c), albeit to a slightly lesser extent than Xist-positive cells. Such global upregulation of X-linked genes has been reported previously in differentiating male mESCs and in male pre- and post-implantation embryos in vivo[30–35]. This process termed X upregulation is thought to have evolved to compensate for the loss of Y-chromosomal genes (Ohno's hypothesis)[36].

**AS analysis of Xist expression patterns**. To quantify Xist in an AS manner, we mapped the sequencing results to the mouse reference genome with masking all SNPs present in the TX1072 cell line, and counted the reads that could be assigned to one or the other allele. Around 4% of all reads could be mapped in an AS manner, detecting ~2000 genes per cell (Supplementary Fig. 4). AS analysis of Xist expression revealed the expected MA pattern after 3–4 days of differentiation, where the vast majority of cells expressed Xist either from the B6 or from the Cast chromosome (Fig. 2a, orange + green). At day 1 and 2 of differentiation by

contrast, a subpopulation expressed Xist from both alleles (Fig. 2a, pink). To quantify the Xist expression patterns, we classified all cells with >5 Xist UMI counts as follows (Supplementary Data 2). Cells were classified as MA, if all counts mapped to the same allele (MA-B6 or MA-Cast, orange, green, Fig. 2a, b). They were labeled as skewed, if more than 80% of Xist counts were assigned to one allele (Fig. 2a, b, light pink) and were defined as biallelic (BA, dark pink), if at least 20% was assigned to each allele.

At day 1 and 2 at least half of Xist-positive cells (>5 UMI counts) expressed both alleles (Fig. 2b, light and dark pink), while at later time points the majority exhibited a MA expression pattern (Fig. 2b, green, orange). These results were confirmed by fluorescent in situ hybridization targeting ribonucleic acid molecules (RNA-FISH), where at day 2 on average 49% of cells exhibited two Xist RNA-clouds of variable size, which were reduced to ~12% and ~5% at days 3 and 4, respectively (Fig. 2c). This BA-to-MA transition between day 2 and 3 was observed independent of the threshold used for AS analysis (Supplementary Fig. 5a). We and others have recently found such transient BA Xist upregulation at the onset of random XCI in vivo[7,8]. Overall, more cells upregulated Xist from the B6 than from the Cast allele (29% vs. 16% at day 4), which is in agreement with the previously reported preferential inactivation of the B6 allele in B6xCast F1 hybrid cells due to differing X-controlling elements (Xce)[37,38]. In addition, the B6 allele appeared to upregulate Xist faster and reached higher levels than the Cast allele (median UMI 31 vs. 16 at day 4) (Fig. 2d). This trend could, however, not be observed in AS bulk RNA-seq data that had been collected in parallel to the single-cell experiment (Supplementary Fig. 5b), suggesting a potential detection bias in the scRNA-seq protocol. In summary, our AS analysis of Xist expression confirmed initial BA upregulation, which was then resolved to a MA state with preferential inactivation of the B6 chromosome.

**Global gene silencing dynamics**. To integrate the analysis of Xist regulation with Xist-induced gene silencing, we next quantified global gene activity of the X chromosome in an AS fashion. First, we calculated for each cell the fraction of X-chromosomal reads (excluding Xist) that mapped to the B6 allele (Fig. 3a). A B6 fraction around 0.5 before differentiation and in differentiating Xist-negative cells reflected equal activity of both X chromosomes. In Xist-expressing cells the distribution broadened over time until two distinct populations became visible at day 4, exhibiting B6 fractions approaching 0 or 1, respectively. This indicated that random X inactivation was initiated, where each cell silenced either the B6 or the Cast X chromosome. Accordingly, the AS X:A ratio decreased on the allele that upregulated Xist (Supplementary Fig. 6a). In the next step, we analyzed the silencing dynamics in individual cells using the concept of RNA velocity[26]. To this end we quantified spliced and unspliced reads for all X-linked genes on each allele separately (Fig. 3b, Supplementary Fig. 4a). The expected XCI trend was observed in both read groups, with the Xist-expressing allele progressively reducing gene activity over time. This observation convinced us to apply the RNA velocity method[26] to predict the future X chromosome activity of each cell during XCI progression. The predicted activity states were then projected onto the first two principal components of the B6 fractions for all X-linked genes (Fig. 3c). The first principle component (PCA 1) separated cells that silenced the Cast (orange) and the B6 chromosome (green) and RNA velocity revealed trajectories toward the silenced states.

We next integrated the X-chromosome-wide silencing analysis with the AS analysis of Xist expression, quantified as the fraction of Xist counts assigned to the B6 allele. We found that Xist

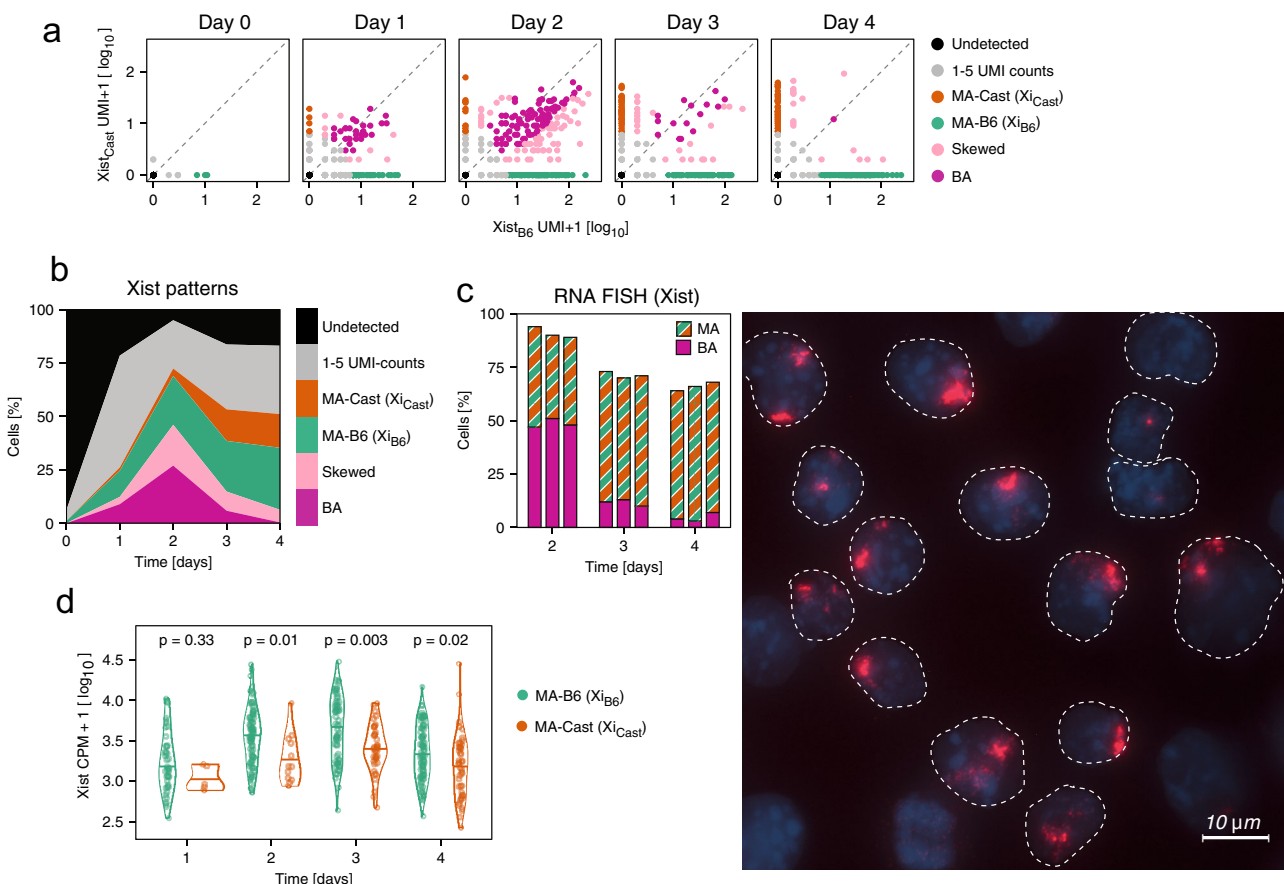

**Fig. 2 Allele-specific analysis of Xist expression. a** Scatter plot showing Xist UMI counts mapping to the B6 and Cast chromosome in individual cells. Cell coloring indicates the Xist pattern classification, where Xist detection from only one allele is termed monoallelic (MA, orange, green). Detection of both chromosomes is termed biallelic (BA, pink), if at least 20% of UMI counts come from each allele, or termed skewed (light pink) if one allele contributes <20% of reads. **b** Relative distribution of Xist patterns as in (**a**) across time. **c** RNA FISH of Xist under the same conditions as in (**a, b**). The bar graph (left) shows quantification of three biological replicates over time (100 cells were counted for each). An example image at day 2 of differentiation (right) is shown, where dotted lines indicate the outline of cell nuclei stained with Dapi (blue). **d** Violin plot comparing Xist allelic expression levels from the B6 (green) and Cast chromosomes (orange) in Xist-MA cells. The horizontal line indicates the median value and *p* values of a Mann–Whitney *U* two-sided test are shown. Source data are provided as a Source Data file.

expression from the B6 allele was associated with a skewing of X-chromosomal gene expression towards the Cast chromosome and vice versa at day 3 and 4 of differentiation, reflecting Xist-induced chromosome-wide gene silencing (Fig. 3d). Moreover, transient BA Xist expression was resolved to a MA state at the same time when chromosome-wide silencing became visible (around day 3). This observation is in line with a role of BA silencing of an X-linked Xist activator in reversing BA Xist upregulation, which we have recently proposed[7]. The extent of gene silencing in cells with BA Xist expression had however remained unknown. Since BA silencing cannot be investigated with the AS approach used above, we instead assessed the X:A ratio in cells with different Xist expression patterns (Fig. 3e). At day 2 and 3 the onset of gene silencing was clearly visible in BA cells and silencing was even more pronounced than in cells with MA Xist expression ($p = 0.45/0.01/0.08$ at day 1/2/3, Mann–Whitney *U* two-sided test), suggesting that gene silencing was indeed initiated at both X chromosomes. In summary, AS scRNA-seq can be used to quantitatively assess the relationship between Xist expression and global gene silencing at the onset of random XCI with single cell resolution. In this way we could show that chromosome-wide gene silencing started around two days after Xist was initially upregulated and coincided with the BA-to-MA transition for Xist.

**Identification of putative Xist regulators**. At all time points we observed that XCI was initiated in a highly heterogeneous fashion, both with respect to Xist expression and X-chromosomal silencing (Figs. 1e and 3a, Supplementary Fig. 6b). We reasoned that we could make use of the variability within the generated single-cell transcriptome data to address the question of how Xist upregulation was triggered at the onset of XCI. We would expect that the responsible Xist activators or repressors should correlate positively or negatively with Xist and early gene silencing across cells. To quantify Xist upregulation we used the normalized Xist UMI-counts per cell, and, as a measure of early silencing, we developed an approach based on the RNA velocity concept[26]. We estimated the change in X-chromosomal gene activity as the ratio between the measured (exonic) expression of a gene and the future expression predicted by RNA velocity. This measure, denoted as $\Delta X$, should increase, when XCI is initiated in a cell. Accordingly, Xist-positive cells showed slightly higher $\Delta X$ values compared to Xist-negative cells at day 2, when XCI was first initiated (Supplementary Fig. 6c).

To identify candidate regulators of XCI initiation, we used two different approaches based on differential expression and correlation analyses, respectively, to ensure robustness of the results (Supplementary Data 3). For the differential expression analysis we compared at each time point cells with low and high

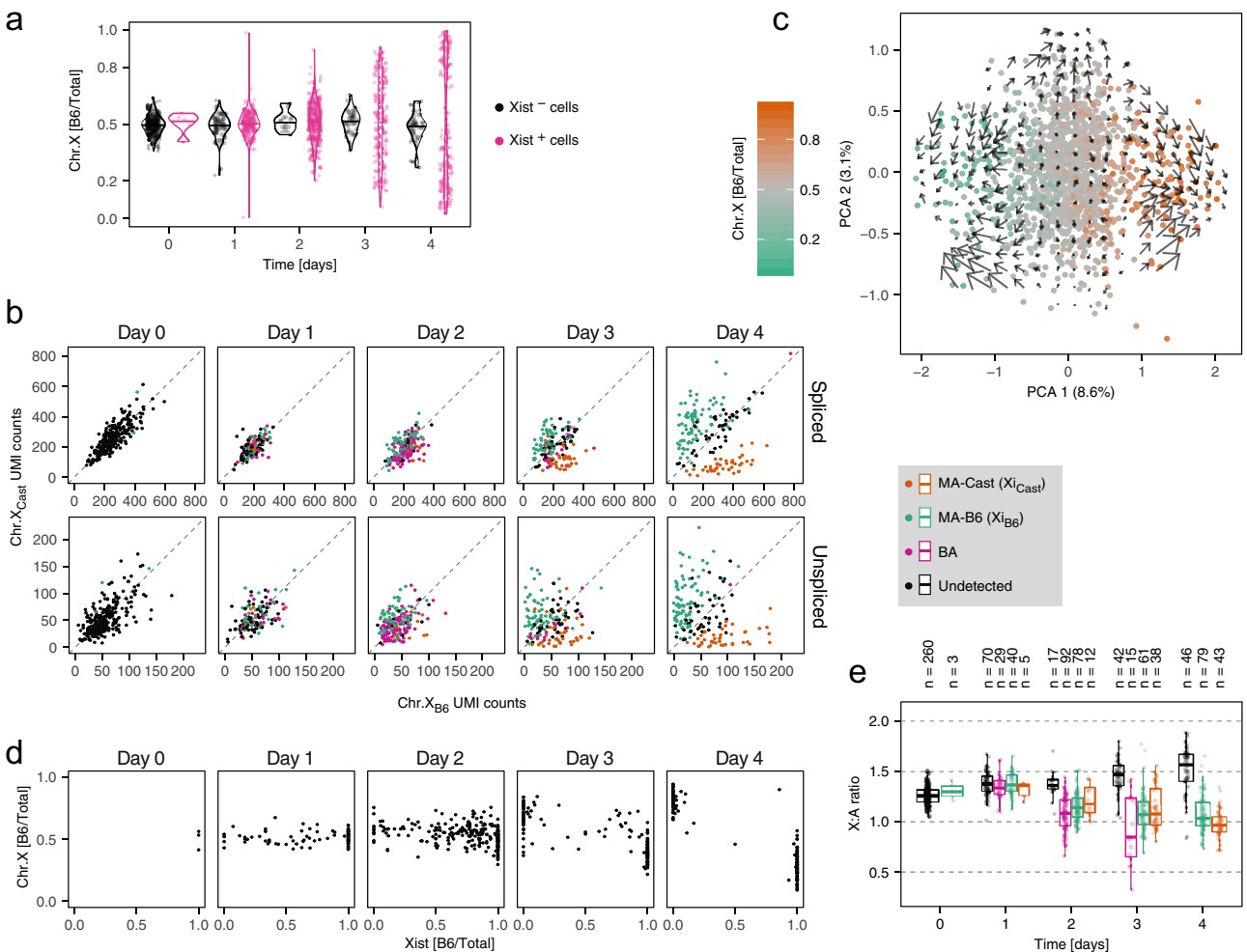

**Fig. 3 Chromosome-wide gene silencing dynamics. a** Violin plot showing the distribution of the allelic expression ratio for the entire X chromosome excluding Xist, as the fraction of X-chromosomal reads mapping to the B6 chromosome, in Xist-positive (pink) and Xist-negative cells (black). Horizontal lines indicate the median values. **b** Scatter plots showing spliced (top) and unspliced (bottom) reads mapping to the X chromosome on the B6 and Cast alleles. Cells are colored by Xist expression pattern as in Fig. 2a and indicated in the gray box on the right. **c** The first two principal components, computed on the allelic expression ratio for all X-linked genes with the color indicating the chromosome-wide allelic fraction. Arrows indicate the predicted change of the X-linked allelic fraction based on RNA velocity analysis. **d** Scatter plot showing the X-chromosomal allelic fraction vs. Xist's allelic fraction, for cells with >5 allele-specific Xist UMI counts. **e** Box plot showing the X:A expression ratio in cells assigned to different Xist classes, as indicated in the gray box above the plot, based on allele-specific mapping as in Fig. 2a. The central mark indicates the median, and the bottom and top edges of the box indicate the first and third quartiles, respectively. The top and bottom whiskers extend the boxes to a maximum of 1.5 times the interquartile range. Dots represent individual cells and cell numbers are indicated on top. Source data are provided as a Source Data file.

Xist counts per million (CPM)-normalized expression and $\Delta X$ values, respectively, defined by $K$-means clustering (Fig. 4a, Supplementary Fig. 7a, for details see "Computational methods"). We then identified differentially expressed genes (DEGs) between the low and high groups at each time point (except day 0) using the MAST method[39] (Fig. 4b, c, Supplementary Fig. 7b, c). For the correlation analysis we calculated for each time point separately the Spearman's correlation coefficient for each detected gene with Xist expression and $\Delta X$ values, respectively (Fig. 4d–f, Supplementary Fig. 7d–f), an approach that has been successfully applied to scRNA-seq data previously[40].

To identify putative regulators of XCI initiation, we focused on day 1 and 2 of differentiation, when silencing was just being initiated (Fig. 3a). Moreover, we excluded X-linked genes exhibiting reduced expression in Xist-high or $\Delta X$-high cells, since their downregulation is likely not the cause, but the consequence of XCI. We then integrated the results from all eight analyses (Xist/$\Delta X$, day1/2, DE/correlation, Supplementary Fig. 8a,

b) and focussed on genes that were significantly differentially expressed or correlated in at least three analyses (Fig. 5).

On day 1, very few putative XCI regulators were identified (Fig. 5), among them the known Xist repressor Nanog[12]. On day 2, Nanog and several other pluripotency-associated factors, such as Prdm14, Esrrb, and Fbxo15 were identified as potential Xist repressors[41–43]. Among the putative activators we found the X-linked kinase Pim2, which cooperates with the Myc transcription factor[44]. Pim2 was identified in 5 out of 8 comparisons and exhibited the highest fold change between Xist and $\Delta X$ high and low groups. Moreover, multiple genes with potential roles in transcriptional regulation or signaling were identified as putative regulators, such as the transcription factor Pou3f1, which is associated with early ESC differentiation, the DNA methyltransferases Dnmt3a and Dnmt3b, the polycomb-like protein Phf19 and the splicing factor Zcrb1[45–47].

Our analysis identified only a subset of the previously proposed regulators. A comprehensive analysis of genes that have been

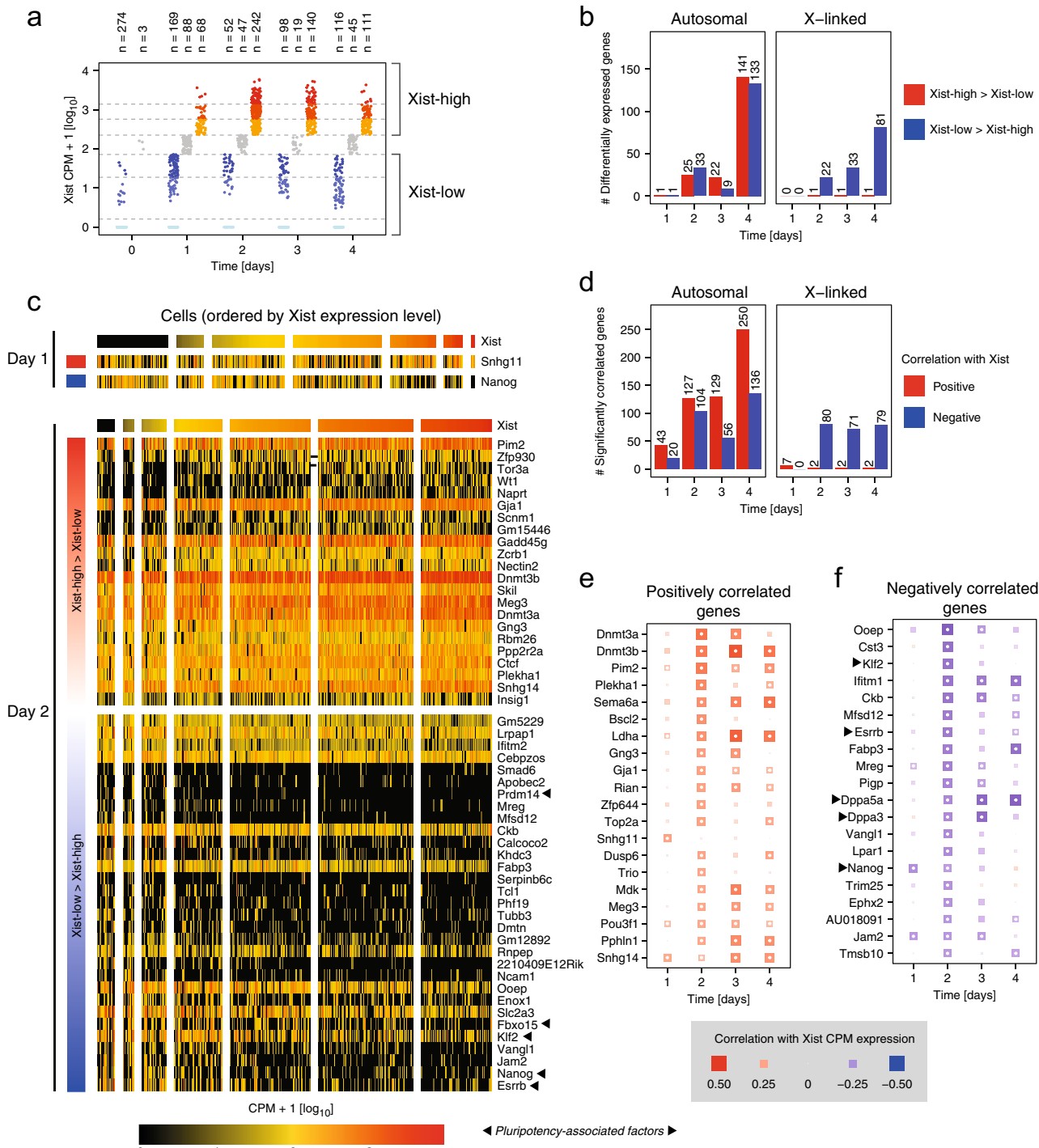

**Fig. 4 Identification of putative Xist regulators. a** Cell classification according to Xist expression levels. The three highest and lowest cell clusters from a K-means (K = 7) clustering were defined as Xist-high (yellow/red) and Xist-low (blue), respectively. **b** Number of differentially expressed genes (DEGs), excluding Xist, between Xist-high and Xist-low cells (the number of cells assigned to the two Xist groups is shown in (**a**)) on autosomes (left, 8880 tested genes) and on the X chromosome (right, 374 tested genes) identified by a $\chi^2$ likelihood ratio test with MAST DE (Benjamini–Hochberg (BH) corrected p value ≤ 0.05). **c** Heatmaps showing expression of DEGs (rows) with absolute fold change between Xist-high and Xist-low cells above 1.5 at day 1 (top) or day 2 (bottom) in single cells (columns). Cells are ordered by Xist expression level and grouped according to the clustering shown in (**a**). Genes are ordered by decreasing fold change. X-linked genes with Xist-low > Xist-high are not shown. **d** Number of genes, excluding Xist, whose expression is positively (red) or negatively (blue) correlated with Xist expression (Spearman's correlation test, BH-corrected p value ≤ 0.05) across cells of the same time point. **e**, **f** Spearman's correlation coefficients with Xist expression for positively correlated genes (**e**) and negatively correlated autosomal genes (**f**), excluding pseudogenes. Top 20 genes that exhibit a significant correlation (Spearman's correlation test, BH-corrected p value ≤ 0.05) at day 1 or 2 are shown, ordered by decreasing absolute correlation coefficient. Size and color indicate the correlation coefficient as indicated. White dots represent significant correlations. Exact p values are provided in Supplementary Data 3. Source data are provided as a Source Data file.

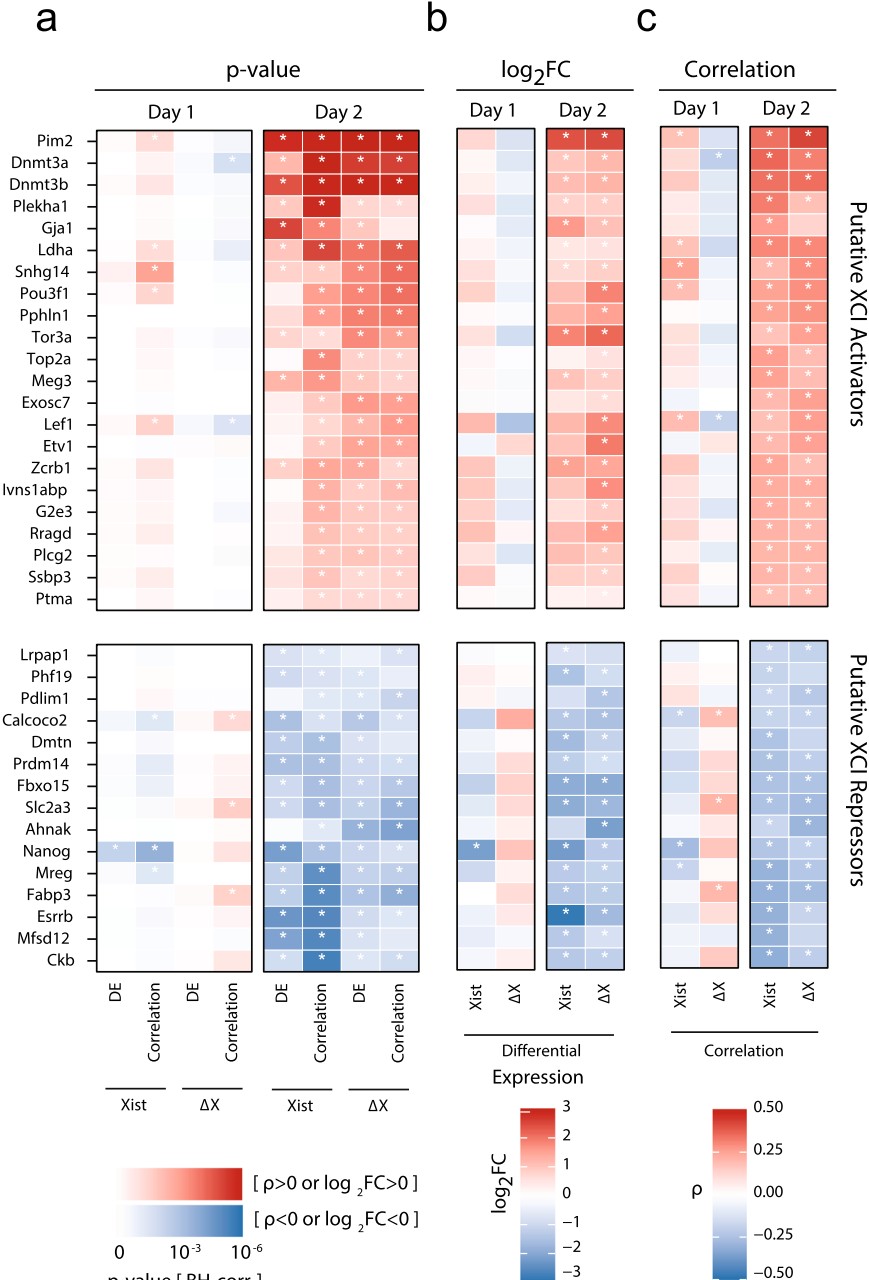

**Fig. 5 Putative regulators of XCI initiation. a–c** Summary of putative regulators of early XCI identified through correlation and differential expression (DE) analyses based on Xist expression and early gene silencing (ΔX) at day 1 and 2 of differentiation. For all genes identified in at least 3 of the 8 analyses (BH-corrected p value ≤ 0.05, asterisks), excluding pseudogenes and X-linked genes with a negative correlation coefficient ρ or log₂FC, the BH-corrected p value in (**a**), the log₂-transformed fold change (log₂FC) in (**b**), and Spearman's correlation coefficient ρ in (**c**) are shown. Source data are provided as a Source Data file.

implicated in Xist regulation before, showed that Nanog, Klf2/4 and Prdm14 were correlated with Xist and early XCI, while other pluripotency factors such as Oct4 (Pou5f1) and Sox2 were not (Supplementary Fig. 8c, d). Taken together, downregulation of naive pluripotency factors, in particular Nanog and upregulation of early differentiation factors, such as Pou3f1 and Dnmt3a/b seemed to initiate XCI. Moreover, the Pim2 kinase showed the strongest association with Xist and early XCI, making it an interesting candidate for further studies.

**Gene- and allele-specific silencing dynamics.** Since our time course experiment was performed in a highly polymorphic cell

line, it also provided the opportunity to ask how genetic variation affected the efficiency or dynamics of gene silencing. When comparing XCI in MA-B6 and MA-Cast cells, we found that the Cast chromosome appeared to be silenced more efficiently (Fig. 6a). The fact that both, Xist (Fig. 2d) and other X-linked genes appeared to be preferentially detected from the B6 chromosome might suggest a technical artifact, such as a mapping bias towards the reference genome (B6). Although we detected a slight tendency for higher expression from the B6 chromosome (1493/1078 autosomal genes with higher expression on B6/Cast, 2288 genes unchanged, $p = 0.002$, Fisher's exact test), the fact that the median allelic ratio was 0.99–1.05 for autosomal genes and for X-

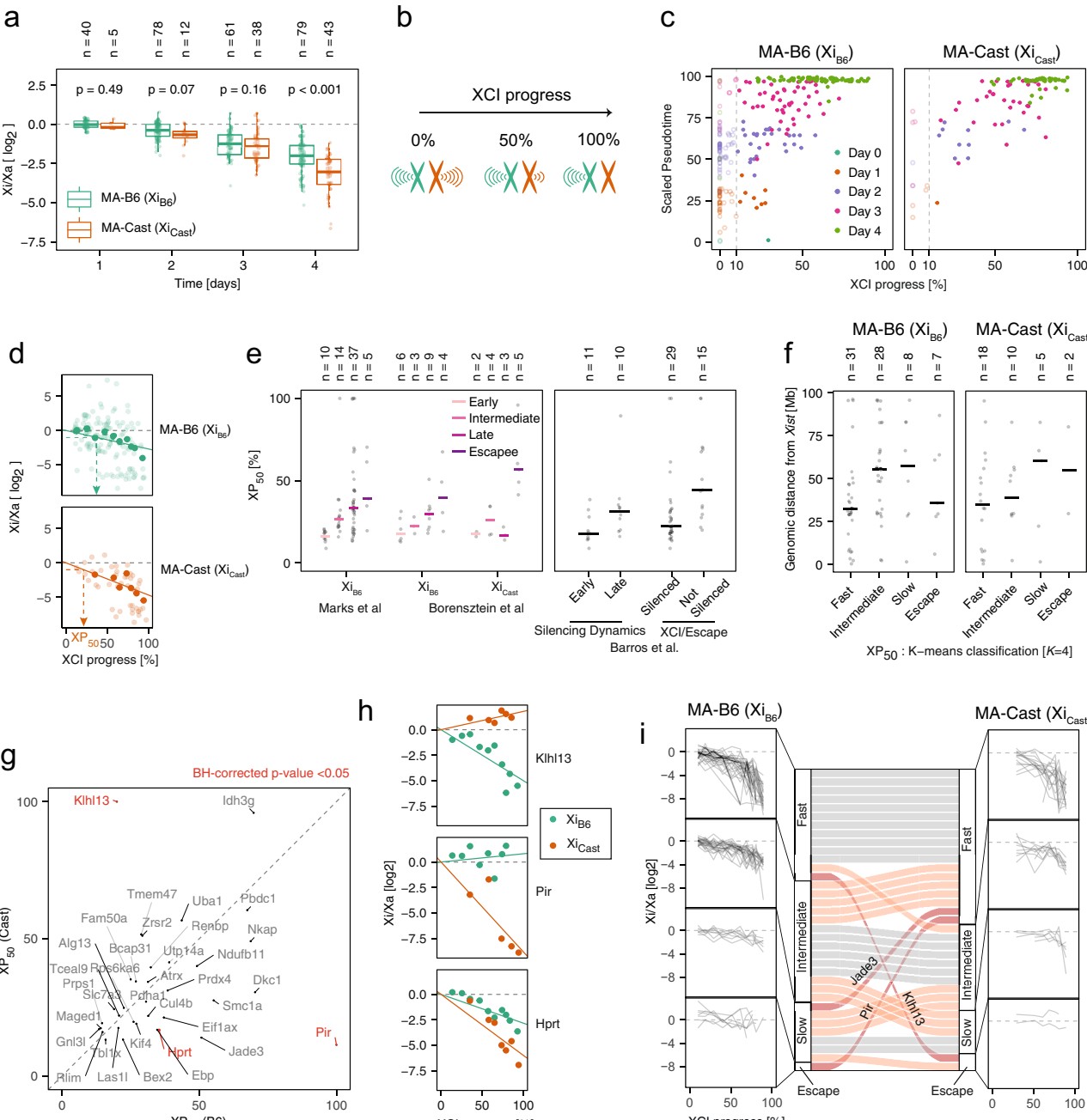

**Fig. 6 Allelic silencing dynamics. a** Xi:Xa expression ratio for cells that inactivate the B6 (green) and Cast chromosome (orange), excluding reads mapping to Xist. The central mark indicates the median, and the bottom and top edges of the box indicate the first and third quartiles, respectively. The top and bottom whiskers extend the boxes to a maximum of 1.5 times the interquartile range. Dots represent individual cells and cell numbers are indicated on top; *p* values of a Mann–Whitney *U* two-sided test are shown. **b** Schematic representation of XCI progress (XP), defined as the percentage of silencing. **c** Comparison of XCI progress with scaled pseudotime as in Fig. 1b for MA-B6 (left) and MA-Cast cells (right). Cells are colored according to measurement time point. Only cells with XP ≥ 10% (vertical dashed line) were included in the differential silencing analysis in (**d–i**). **d** Estimation of allele-specific silencing dynamics shown for an example gene (Eif1ax). Transparent dots indicate individual cells and solid dots the binned values for cells with similar XCI progress. The solid line shows the log-linear fit, used to estimate the $XP_{50}$ value (dashed arrows), which describes the global XCI progress at which a given gene is silenced by 50%. **e** Comparison of the estimated $XP_{50}$ values with previously determined silencing classes[18,19,32]. Dots represent individual genes and the horizontal bars show the median value. The number of genes in each group are given on top. **f** Genomic distance from the *Xist* gene for genes grouped according to their $XP_{50}$ values ($K$-means clustering with $K = 4$) on the B6 (left, 74 genes) and Cast chromosomes (right, 35 genes), respectively. Dots represent individual genes and the horizontal bars show the median value. The number of genes in each group is given on top. **g, h** Comparison of $XP_{50}$ values estimated for the B6 and Cast chromosomes. Genes with significantly different silencing dynamics (ANOVA *F* test: BH-corrected *p* value ≤ 0.05) are colored in red and shown in (**h**). **i** *K*-means clustering of genes according to their allelic $XP_{50}$ value as in (**f**). Connecting bars in the center compare classification for genes that could be analyzed on both chromosomes (35 genes). Gray bars indicate genes that were assigned to the same cluster for the two alleles, light red indicates genes that were classified in neighboring clusters and dark red are genes that are part of more distant clusters. Source data are provided as a Source Data file.

linked genes in Xist-negative cells did not provide an indication for a strong mapping bias (Supplementary Fig. 9a, b). We thus concluded that genetic variation between the B6 and Cast alleles seemed to result in Xist being upregulated with higher probability from the B6 X chromosome, while subsequent silencing was induced more rapidly on the Cast allele.

To be able to compare silencing efficiency between alleles of individual genes, we developed an approach to quantify silencing of each gene relative to the rest of the chromosome. In this way we aimed at normalizing for the observed global differences in X-inactivation dynamics. To this end we quantified the "XCI progress" (XP) of each cell with MA Xist expression as the percentage of silencing of the inactive X chromosome (Fig. 6b, see computational methods for details). Although XP correlated with the time point when cells were collected (Spearman's correlation coefficient $\rho = 0.73$, $p < 0.001$) and with the pseudotime estimated based on the global transcriptome (Spearman's correlation coefficient $\rho = 0.76$, $p < 0.001$), XCI appeared highly variable even within cells collected at the same time point or associated with a similar pseudotime (Fig. 6c). To quantify the silencing state for each gene relative to the rest of the chromosome, we grouped cells into 10 bins according to their XP and calculated a lumped Xi:Xa ratio across cells in each bin for each gene (Fig. 6d, solid dots). We then fitted a log-linear model to the Xi:Xa ratio across bins, separately for cells silencing the B6 or Cast chromosomes (Fig. 6d, lines). In this way we estimated an allele- and gene-specific XCI half time, corresponding to the global XP where a gene was silenced by 50% ($XP_{50}$) (Fig. 6d).

To validate this approach we compared the $XP_{50}$ values with results from previous studies, where genes had been classified according to their silencing dynamics on the B6 chromosome (or the closely related 129 allele), based on bulk sequencing of mature or nascent RNA[18,19]. Moreover, we compared the estimated $XP_{50}$ values with a previous classification based on scRNA-seq measurements in pre-implantation mouse embryos, where an imprinted form of XCI occurs, that had been performed for both alleles[32]. Our estimated $XP_{50}$ values were in good agreement with these classifications (Fig. 6e). In addition, we used K-means clustering to group all genes into four categories according to their $XP_{50}$ values (fast, intermediate, slow, and escape), separately for each allele. Analyzing the distance from the Xist locus for genes in each category confirmed the previously described trend that silencing tends to be faster in closer proximity to Xist[18,19] (Fig. 6f).

In the next step we compared the estimated $XP_{50}$ values between the two alleles for each gene (Fig. 6g). As expected the majority of genes exhibited similar silencing dynamics, since we had normalized for global silencing differences through the XP approach. We found a subset of genes (Klhl13, Pir, and Hprt), which were silenced with significantly different dynamics on the two alleles (ANOVA F test: BH-corrected $p$ value < 0.05, Fig. 6h). While Klhl13 appeared to escape on the Cast chromosome and was silenced on B6, Pir, and Hprt were silenced more slowly on the B6 chromosome. These three genes were consistently identified, even when varying the analysis parameters (Supplementary Fig. 9c). Finally we analyzed, which genes were clustered into different silencing categories on the two alleles (Fig. 6i). Here, 3 out of 35 genes were assigned to the fast category on one allele and to the slow or escaping group on the other (Klhl13, Pir, and Jade3), suggesting again the existence of AS escape genes, such as Klhl13 and Pir. Taken together, our results show that overall, the Cast chromosome is silenced faster than the B6 allele. In addition, susceptibility to Xist-mediated gene silencing appears to be altered by genetic variation for a subset of genes. To further validate these findings we aimed to estimate AS silencing dynamics with an orthogonal experimental approach.

**Experimental validation of AS silencing dynamics.** To assess XCI dynamics on the B6 and Cast chromosomes independently, we generated two mESC lines, where the X-inactivation center (Xic), which encompasses the Xist gene, was deleted on either one or the other allele, and named them TXΔXic_B6 and TXΔXic_Cast, respectively (Fig. 7a). To this end, a ~800 kb region around the Xist locus was deleted through Cas9-mediated genome editing (Supplementary Fig. 10). Upon differentiation these cell lines underwent nonrandom XCI (Fig. 7a, Supplementary Fig. 11a), allowing us to use bulk assays to measure gene silencing. We verified that both cell lines upregulated Xist with comparable dynamics and that the fraction of Xist-expressing cells was similar (Fig. 7b, Supplementary Fig. 11b, c). We quantified the relative allelic expression over a 4-day differentiation time course through AS bulk RNA-sequencing and pyrosequencing. Overall, the Xi:Xa ratio decreased more strongly in the TXΔXic_B6 line compared to TXΔXic_Cast cells, when assessed for the entire X chromosome by RNA-seq (Fig. 7c, Supplementary Fig. 11d). This trend was confirmed, when quantifying the allelic expression of five X-linked genes with comparable $XP_{50}$ values on both alleles (Renbp, Atrx, Cul4b, Prdx4, and Rlim) through pyrosequencing, which performs quantitative sequencing over individual SNPs on cDNA (Supplementary Fig. 11e, f).

In addition we analyzed all three genes that were found to be differentially silenced between the alleles in our scRNA-seq analysis (Fig. 6g, h). For Klhl13, which we had found to escape on the Cast chromosome, while being silenced on the B6 allele, we indeed observed a strong decrease of the Xi:Xa ratio on the B6 chromosome and even an increase in the line that silenced the Cast allele (Fig. 7d, Supplementary Fig. 11g). The two other genes we tested, Pir and Hprt, had been found to be silenced faster on the Cast allele and we could indeed confirm that, at later time points, the Xi:Xa ratio appeared to be reduced more strongly in the line that inactivated the Cast chromosome (Fig. 7d, Supplementary Fig. 11g). It must be noted, however, that we cannot distinguish at this point, whether these differences arise only from the overall increased efficiency of XCI on the Cast allele, or whether indeed additional gene-specific effects also contribute. Taken together, we could confirm faster silencing of the Cast allele in an independent experimental approach and we could clearly validate Klhl13 as an AS escape gene.

## Discussion

We have profiled multiple steps governing the onset of random X inactivation with temporal and allelic resolution through single-cell RNA sequencing. By adapting single-cell transcriptome analysis approaches, including recently developed concepts like RNA velocity, to an AS process such as random XCI, we could draw a detailed picture of the different steps occurring at the onset of X inactivation. In this way we were able to answer several open questions by (1) dissecting the dynamics of XCI along developmental progression with allelic resolution and (2) quantifying the expression heterogeneity of the main genes involved in the XCI process, their regulatory relationships and their dynamics. Due to an efficient library preparation protocol and sufficient sequencing depth we could detect a median of ~120,000 mRNA molecules per cell, which is significantly more than other UMI-based methods[48]. This allowed us to quantify allelic expression for individual genes, including Xist and genes that are subject to XCI. We identified different Xist patterns at different stages of the XCI process and their associated gene silencing state. In addition, we exploited the heterogeneous nature of XCI initiation to identify putative regulators of Xist. Finally, our AS analysis revealed marked differences between the B6 and Cast X

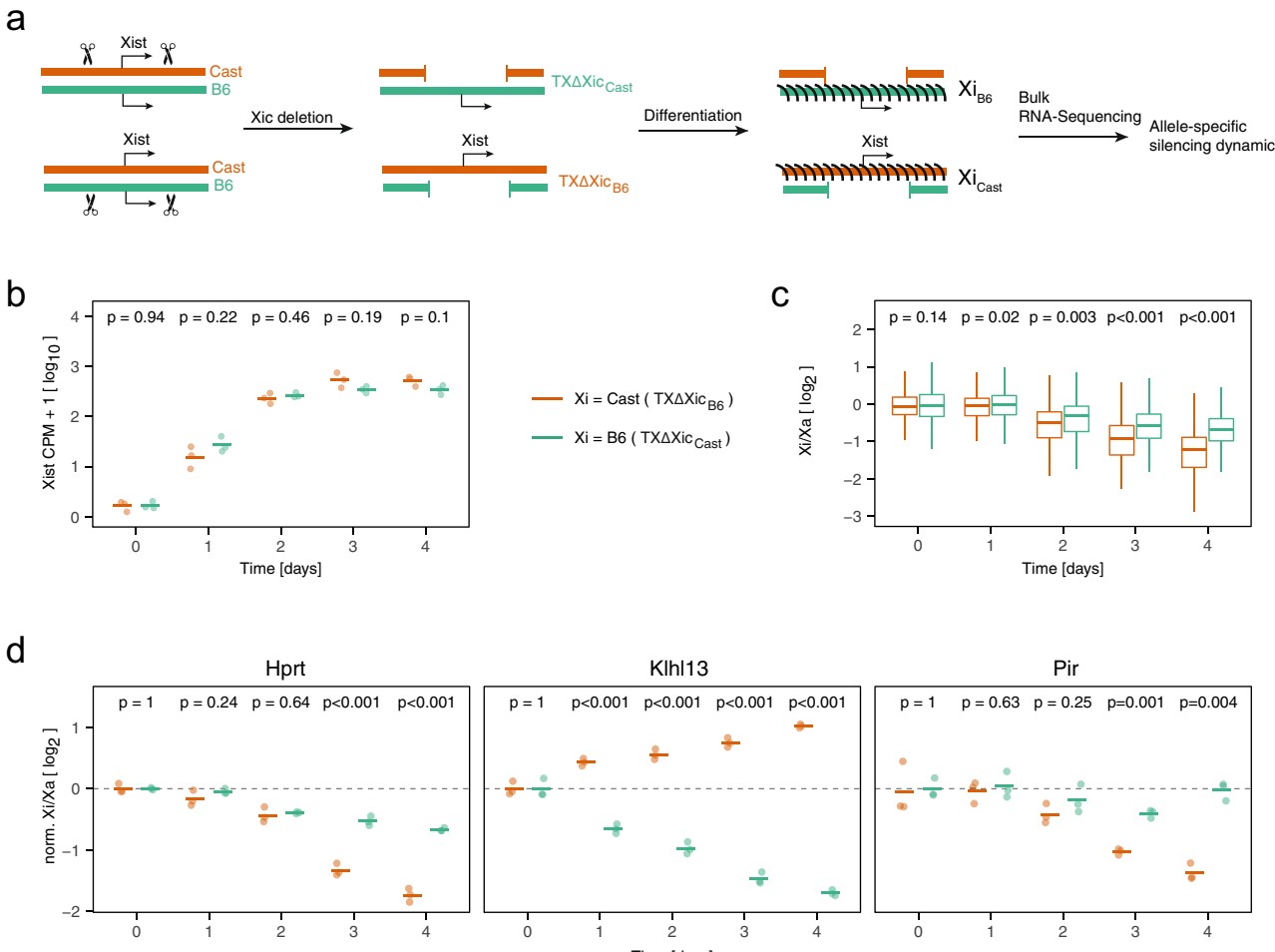

**Fig. 7 Experimental validation of differential silencing dynamics. a** To measure allele-specific silencing dynamics, the X inactivation center (Xic), which contains the *Xist* locus, was deleted on either the Cast (top) or on the B6 allele (bottom). Upon differentiation, the entire cell population will thus initiate XCI on the B6 and Cast allele, respectively, allowing quantification of allele-specific silencing dynamics by bulk RNA-sequencing, as shown in (**b-d**). **b-d** RNA-seq differentiation time course of the cell lines shown in (**a**). **b** Xist expression. Horizontal bars represent the mean of three biological replicates, dots the individual measurements, significance (*p* values) of the difference between the two cell lines according to a two-sample two-sided unpaired Student's *t* test is indicated. **c** Xi:Xa expression ratios for all X-linked genes outside the deleted region ($n = 660$ genes) averaged across $n = 3$ biological replicates. Significance (*p* values) of the difference between the two cell lines according to a two-sample two-sided paired Student's *t* test is indicated. The central mark indicates the median, and the bottom and top edges of the box indicate the first and third quartiles, respectively. The top and bottom whiskers extend the boxes to a maximum of 1.5 times the interquartile range. Outliers are not shown. **d** Xi:Xa expression ratios for differentially silenced genes, as identified in Fig. 6, normalized to the average ratio on the pre-XCI state (day 0, dashed line). Horizontal bars represent the mean of three biological replicates, dots the individual measurements. In addition, *p* values of a two-sample unpaired two-sided Student's *t* test comparing the two cell lines are shown. Source data are provided as a Source Data file.Source Data

chromosomes, suggesting that *cis*-acting genetic variants modulate both, Xist expression and gene silencing dynamics.

We used a strand-specific, 3′-end counting scRNA-seq protocol. This enabled a detailed quantitative analysis of Xist expression, because it allowed us to clearly distinguish Xist from its antisense transcript Tsix[49], in contrast to previous studies, where Xist had been analyzed in single cells with unstranded full-length scRNA-seq techniques[22,23]. Moreover, Xist seemed to be detected more efficiently in the 3′-end protocol used here, potentially due to inefficient full-length amplification of very long RNAs such as Xist. The downside of 3′-end approaches is that only a subset of SNPs close to the 3′-end can be used for AS analyses, leading to a reduced number of genes with allelic resolution, compared to full-length methods. Although the protocol generally detected the 3′-end of most genes, since reverse transcription (RT) was initiated from the polyA tail, this was not the case for Xist, suggesting that Xist's 3′-end is inaccessible during the RT reaction. This effect is

not specific to the protocol used here, since we observed a similar pattern in data generated with CEL-seq2[29]. Moreover, we and others have failed previously to amplify Xist in single-cell mRNA libraries, both in ES cells in vitro and in blastocyst stage embryos in vivo, which was at the time circumvented by addition of an Xist-specific RT primer[21,50]. Xist's 3′-end thus seems to be covered by a structure that can resist the mild denaturation conditions in scRNA-seq protocols. This poses a general question of how many other RNAs, maybe specifically nuclear lncRNAs might escape detection in scRNA-seq experiments.

Although reverse transcription could not be initiated from Xist's polyA tail, a genomically encoded polyA stretch served as a priming site instead, allowing us to nevertheless quantify Xist expression in an AS fashion. Such polyA stretches have also been proposed to underlie detection of intronic reads used for RNA velocity analyses in 3′-end scRNA-seq data sets[26]. At the early time points we identified a subset of cells that expressed Xist from

both chromosomes, which constituted up to ~50% of the population. We and others have previously observed such transient BA Xist upregulation by RNA FISH, both in differentiating mESCs and in mouse embryos in vivo[7,8,51]. We found that BA Xist upregulation actually initiated gene silencing and that it was resolved, when chromosome-wide silencing became visible. These observations are partially reminiscent of early human development, where a stage with BA Xist expression and partial silencing ("dampening") has been described[52,53]. In humans, however, this state persists for several days, but must eventually also be resolved to MA Xist expression and chromosome-wide silencing, as observed in somatic cells. These observations are in line with the idea we have previously proposed that complete silencing of an essential X-linked Xist activator might lead to Xist downregulation in cells where XCI has been initiated on both chromosomes[7]. Accordingly, we find that Rnf12/Rlim, which functions as such an activator[15,54], is indeed rapidly silenced, leading even to a strong negative correlation with Xist at day 2 (Supplementary Fig. 8d). Our data thus lends strong support to the stochastic model of XCI, where initial Xist upregulation occurs independently on each allele with a low probability, and a silencing-mediated negative feedback loop then ensures that only a MA state can be maintained[10].

The stochastic nature of Xist upregulation was also supported by our finding that Xist expression was heterogeneous across cells and the correlation with its regulators was rather weak (absolute correlation coefficient < 0.32). While two previous studies had attempted to identify XCI regulators by analyzing correlation with X-chromosomal gene activity[22,31], our analysis allows identification of putative early regulators of Xist. When Xist was first upregulated, at day 1 of differentiation, very few differences could be detected between the transcriptomes of cells expressing Xist and those that had not yet initiated XCI. A notable exception was Nanog, which was decreased in cells initiating XCI. This suggests that initial Xist upregulation is linked to downregulation of naive pluripotency factors, such as Nanog, independently of Oct4 and Sox2, which have also been proposed as developmental Xist regulators[12,13]. In addition, we found several putative XCI activators, including the X-linked kinase Pim2, which warrants further investigation in the future.

The highly polymorphic cell line we used here allowed an AS analysis, which is required to quantify gene expression from each X chromosome individually. We found significant differences between the two X chromosomes, suggesting that cis-acting genetic variation modulates the X inactivation process at multiple levels. First, we observed the long-known Xce effect leading to preferential inactivation of the B6 X chromosome, which has been mapped to a large genomic region centromeric to Xist[1,38,55]. However, it was the Cast allele that silenced genes more efficiently. This observation suggests that polymorphisms within the Xist RNA might reduce silencing efficiency of Xist in the B6 strain, for example by altering binding affinity for silencing factors such as Spen[6]. An intriguing hypothesis is that such reduced silencing efficiency might have been evolutionarily compensated by faster Xist upregulation at the B6 X chromosome, ultimately leading to the long-known Xce effect in hybrid mouse embryos.

Since XCI is a chromosome-wide process it can be quantified by scRNA-seq in a fairly robust manner despite the limited sensitivity of the method. Due to the high quality of the dataset we have generated we could in addition compare silencing of individual genes between alleles. We found that a subset of genes were silenced with different dynamics, in agreement with a similar observation made during imprinted XCI in pre-implantation embryos[32]. Although substantial heterogeneity with respect to escape from XCI has been reported in single human

fibroblasts[56], no systematic analysis of putative genetic determinants of this heterogeneity has been performed to date. An extreme scenario with complete escape from XCI specifically on the Cast chromosome was observed for Klhl13 and confirmed in an independent experiment. We have recently identified Klhl13 as an X-linked differentiation inhibitor[57]. Its silencing might help to release a differentiation block imposed by the presence of two active X chromosomes[21,58]. Escape of Klhl13 in the Cast strain might, therefore, have evolved to compensate for the overall faster silencing of the X chromosome in that genetic background, which might otherwise result in a too fast release of the differentiation block. In the future, strain-specific escapees such as Klhl13 will potentially allow us to identify escape-promoting cis-acting genetic elements to better understand the principles that can protect a gene from inactivation. Escape mechanisms are a key unanswered question in XCI research and will, once elucidated, be an important contribution to understanding epigenetic gene regulation.

The strategies we use here to profile X inactivation in the endogenous setting of random XCI will be valuable to investigate XCI status in other contexts, in particular those that are not amenable to genetic engineering. XCI and escape can, for example, be assessed in primary human cells, making use of naturally occurring genetic variation. This approach has been applied successfully to identify escape genes in human lymphoid cells, for which the genomic sequence was known[59]. However, SNP positions can likely be identified from the scRNA-seq data itself through variant-calling methods and can be phased based on expression correlation across cells. Although humans carry less heterozygous SNPs than the hybrid mouse strain used in our study, which might result in a smaller number of genes with allelic information, the global XCI status can probably still be inferred. Combined with the power of single-cell genomics to cluster cell types, we can then assess the stability of the inactive X chromosome across different tissues and cell subsets. Such information will be indispensable to understand how variability in XCI contributes to sexual dimorphic traits and to differential disease susceptibility between the sexes such as the higher prevalence of autoimmune diseases in women[60].

## Methods

### Experimental procedures

*Cell lines.* The female TX1072 cell line (clone A3) is a F1 hybrid ESC line derived from a cross between the C57BL/6 (B6) and CAST/EiJ (Cast) mouse strains that carries a doxycycline-responsive promoter in front of the *Xist* gene on the B6 chromosome and an rtTA insertion in the *Rosa26* locus[21]. TxdXic_A1 and TxdXic_A6 lines were generated by deleting a 773 kb region around the *Xist* locus. TxdXic_A1 carries the deletion on the B6 chromosome (chrX:103,182,701–103,955,531, mm10) and was thus named TXΔXic$_{B6}$ and TxdXic_A6 on the Cast allele (chrX:103,182,257–103,955,698, mm10) and was named TXΔXic$_{Cast}$. To generate these cell lines, a total of four guides flanking the region to be deleted (two on each side) were cloned into the pX459-v2 vector (Addgene #62988)[61]. Four separate transfections were performed, each combining one guide upstream and downstream of the targeted region with a single-stranded repair oligo with 50 bp homology arms flanking each cut site (IDT). Further details and sequences are provided in Supplementary Data 4. In each transfection, $2 \times 10^6$ TX1072 mESCs were electroporated with 2 µg of each guide plasmid and 30 pmol of the single-stranded repair oligo using the P3 Primary Cell 4D-Nucleofector X Kit (V4XP-3024) with the Amaxa 4D Nucleofector system (Lonza, program CP-106) and plated on gelatin-coated 10 cm dishes. Starting 30 h after transfection, cells were selected with puromycin (1 µg/ml) for 16 h. After 4 days equal cell numbers from all 4 transfections were pooled and plated at a density of 2–3 cells/well in gelatin-coated 96-well plates. Genomic DNA was extracted and screened for the presence of the deletion with primers VM21 and VM24. Cells from wells showing the presence of the deletion were seeded at clonal density (100 cells/cm²) in 10 cm plates. Individual clones were picked and screened for the presence of the deletion and the wildtype allele and the absence of inversions or duplications of the targeted region. The genotyping strategy is shown in Supplementary Fig. 10. For all PCR reactions the Phusion HotStart Flex DNA Polymerase (NEB) was used with an annealing temperature of 55 °C, an elongation time of 45 s and 35 cycles. Primer sequences are listed in Supplementary Data 4. To determine which allele carried

the deletion, the PCR products were sequenced and annotated SNPs between Cast and B6 alleles were used to assign the deleted and the wild-type allele. Correct karyotype of clones was verified with metaphase spreads.

*mESC culture and differentiation.* Cells were grown on gelatin-coated flasks in serum-containing ES cell medium (DMEM (Sigma), 15% ESC-grade fetal bovine serum (FBS) (PanBiotech), 0.1 mM $\beta$-mercaptoethanol, 1000 U/ml leukemia inhibitory factor (LIF, Millipore)), supplemented with 2i (3 μM Gsk3 inhibitor CT-99021, 1 μM MEK inhibitor PD0325901, Axon). Differentiation was induced by 2i/LIF withdrawal in DMEM supplemented with 10% FBS and 0.1 mM $\beta$-mercaptoethanol at a density of $1.5 \times 10^4$ cells/cm$^2$ in fibronectin-coated (10 μg/ml) tissue culture plates.

*scRNA-seq.* Single-cell RNA-seq libraries were prepared with the C1-HT mRNA-seq v2 protocol according to the manufacturer's recommendations (Fluidigm). Cells were rinsed thoroughly with phosphate-buffered saline (PBS), trypsinized for 7 min and resuspended in the respective growth medium at a concentration of 400 cells/μl. Thirty microlitre cell suspension was diluted with 20 μl of suspension reagent (Fluidigm) and 10 μl of the dilution was loaded into one compartment of a single-cell mRNA Seq HT integrated fluidic circuit (IFC) 10–17 μm. A different cell type was loaded into the other compartment, which is not analyzed in this study. Cell viability staining was performed on the IFC using the LIVE/DEAD viability/Cytotoxicity Kit (Thermofisher) with 1 μM Ethidium and 0.05 μM Calcein. IFC loading and life/dead staining was analyzed with automated image acquisition using a Zeiss CellDiscoverer microscope (Zeiss) with a 20× objective. During the lysis step ERCC Spike-in Mix 1 (Thermofisher) was added with a final dilution of 1:200,000. Lysis, reverse transcription, and cDNA amplification was performed on the C1 machine. cDNA pools were quantified by Qubit and Bioanalyzer HS. Around 2.25 ng of each pool were subjected to tagmentation and library preparation using the NexteraXT library preparation kit according to the C1-HT protocol. All pools were mixed in equal proportions and quantified with KAPA Library Quant-Kit. The libraries were sequenced on a HiSeq2500 instrument (Illumina) with asymmetric read length, either in High Output (Read1: 13 bp, Index read: 8 pb, Read2: 48 bp) or in Rapid Run mode (Read1: 16 bp, Index read: 8 pb, Read2: 36 bp), with 10 pM loading concentration and 5% PhiX.

*RNA extraction, reverse transcription, and qPCR.* For gene expression profiling, cells were lysed directly in the plate by adding 1 ml of Trizol (Invitrogen). RNA was isolated using the Direct-Zol RNA Miniprep Kit (Zymo Research) following the manufacturer's instructions. For pyrosequencing, DNAse digest was performed using Turbo DNA free kit (Ambion). One microgram RNA was reverse transcribed using Superscript III Reverse Transcriptase (Invitrogen) and expression levels were quantified using Power SYBR$^{TM}$ Green PCR Master Mix (4368702, Thermo Fisher) and normalized to Rrm2 and Arpo. Primer sequences are listed in Supplementary Data 4.

*Bulk RNA-sequencing.* In parallel to the sc-RNA-seq experiment, bulk RNA-seq was performed from the same cell population (TX1072, replicate 1) and for two more biological replicates (replicate 2 + 3). RNA-seq libraries were generated using the Tru-Seq Stranded Total RNA library preparation kit (Illumina) with 1 μg starting material for rRNA-depletion and amplified with 15 Cycles of PCR. Libraries were sequenced 2× 50 bp on a HiSeq 2500 with 1% PhiX spike-in, which generated ~50 Mio fragments per sample. For the TXΔXic cell lines, libraries were generated with the KAPA-RNA Hyper Prep-Kit with RiboErase (Roche) following the protocol, with 500 ng total RNA used for rRNA-depletion. For undifferentiated TXΔXic$_{Cast}$ samples fragmentation was adjusted (85 °C/5 min instead of 94 °C/8 min) due to RNA degradation. Nextflex unique dual-index-adapters (PerkinElmer) were used and the final library was PCR-amplified with 10 cycles. Libraries were sequenced 2× 100 bp on a NovaSeq6000 with 1% PhiX spike-in, which generated ~50 Mio fragments per sample.

*Pyrosequencing.* To quantify relative allelic expression for individual genes, an amplicon containing a SNP was amplified by PCR from cDNA using GoTaq Flexi G2 (Promega) with 2.5 mM MgCl$_2$ or Hot Star Taq (Qiagen) for 40 cycles. The PCR product was sequenced using the Pyromark Q24 system (Qiagen). Assay details are given in Supplementary Data 4.

*RNA FISH.* RNA FISH for Xist and another X-linked gene, Huwe1 was performed using Stellaris FISH probes (Biosearch Technologies). Probe details can be found in Supplementary Data 4. Cells were dissociated using Accutase (Invitrogen) and adsorbed onto coverslips (#1.5, 1 mm) coated with Poly-L-Lysine (Sigma) for 5 min. Cells were fixed with 3% paraformaldehyde in PBS for 10 min at room temperature (18–24 °C) and permeabilized for 5 min on ice in PBS containing 0.5% Triton X-100 and 2 mM Ribonucleoside Vanadyl complex (New England Biolabs). Coverslips were preserved in 70% EtOH at −20 °C. Prior to FISH, coverslips were incubated for 5 min in Stellaris RNA FISH Wash Buffer A (Biosearch Technologies), followed by hybridization overnight at 37 °C with 250 nM of each FISH probe in 50 μl Stellaris RNA FISH Hybridization Buffer (Biosearch Technologies) containing 10% formamide. Coverslips were washed twice for

30 min at 37 °C with Stellaris RNA FISH Wash Buffer A (Biosearch Technologies), with 0.2 mg/ml Dapi being added to the second wash. Prior to mounting with Vectashield mounting medium coverslips were washed with 2× SSC at room temperature for 5 min. Images were acquired using a widefield Z1 Observer microscope (Zeiss) using a 100× objective. The intronic signal of Huwe1 was used in combination with Xist to estimate the percentage of XO cells in the population, which was maximally 5%.

## Computational methods

*Alignment and gene quantification. Preprocessing of scRNA-seq data.* The C1-HT protocol uses a dual barcoding strategy, where read R1 contains a custom barcode (position 1–6, cell barcode) and a UMI (position 7–11, UMI), read R2 maps to the cDNA sequence and read R3 contains a Nextera (row) barcode. After demultiplexing using the Nextera barcode, read R2 was aligned with STAR (v.2.5.2b)[62] to the mouse genome genome (mm10) and all 96 ERCC spike-in sequences, allowing for unique alignments with a maximum of two mismatches. SNPsplit (v.0.3.2)[63] was used to N-mask the genome using high confidence SNPs, confirmed to be present in the TX1072 cell line based on ChIP input data[19].

The Drop-seq pipeline[64] was used to extract both cell barcode and UMI from read R1 and to quantify gene expression. In detail, reads were demultiplexed using the cell barcode, and molecules per gene were counted as the number of unique exon-overlapping UMI barcodes using the mm10 Ensembl gene annotation. For RNA velocity analyses unique UMI barcodes aligning only to exonic regions (spliced) or overlapping with intronic regions (unspliced) were used. For AS quantification each read overlapping a SNP position was assigned to its parental genome using SNPsplit (v.0.3.2)[63] and the number of unique UMI barcodes assigned to either parental genome were counted for each gene (Supplementary Fig. 4).

*Preprocessing of bulk RNA-seq data.* For both TX1072 and TXΔXic cell lines paired-end reads were aligned with STAR (v.2.5.2b)[62] as described above. The *featureCounts* R function from Rsubread (v.1.34.7)[65] was used to quantify the expression of each gene as the number of uniquely aligned exon-overlapping reads using the mm10 Ensembl gene annotation. AS quantification was performed as described above. Furthermore, Xist (ENSMUSG00000086503) AS gene expression was also quantified using only two SNPs on its 5′-end (chrX:103,482,240 and chrX:103,482,895, mm10) (Supplementary Fig. 5b).

*Cell filtering.* To remove empty wells, dead and low-quality cells, several cell filtering steps were performed (Supplementary Data 1). Capture sites without a cell or with multiple cells were identified based on manual inspection based on brightfield and fluorescence imaging (live/dead stain) of the IFC and removed from subsequent analyses. Moreover, dead cells, cells with low number of reads, low number of transcripts or low number of expressed genes, as well as cells with high percentage of mitochondrial DNA or ERCC spike-in reads were removed from the analysis (Supplementary Fig. 1b). To identify dead cells, the fluorescent signal of the dead stain (Ethidium) was quantified as the integrated intensity within a rectangle of constant size, manually centered around each capture site using ZEN v2.3 software (Zeiss). Thresholds were set as the threefold median-absolute-deviation above or below the median of the respective variable $x$ (e.g., number of reads) for the removal of cells with high and low signal, respectively: $\text{median}(x) \pm 3 \times \text{median}(|x - \text{median}(x)|)$. Finally, cells that did not express Xist and where >80% of X-linked reads mapped to the same allele were assumed to have lost one X chromosome (XO). To identify such cells, the X-chromosomal ratio was defined for each cell $c$ as

$$R_c = \frac{\sum\limits_{g} e_{g,c}^{B6}}{\sum\limits_{g}(e_{g,c}^{B6} + e_{g,c}^{Cast})}, \tag{1}$$

where $g$ denotes a gene on chromosome X, and $e_{g,c}^{B6}$, $e_{g,c}^{Cast}$ are the numbers of AS molecules overlapping exons of gene $g$ in cell $c$ on the B6 and Cast allele, respectively. Cells with $R_c \geq 0.8$ or $R_c \leq 0.2$, which did not express Xist were assumed to be XO and were removed from the analysis (Supplementary Fig. 1c).

*Gene filtering.* In order to analyze only genes with sufficient detection rate across cells, genes that were detected in <20% of all cells were excluded from the analysis. Gene filtering was performed separately for the AS and not-AS analysis. Furthermore, genes with a strong skewing toward a single allele where >90% of reads detected in all cells mapped to the same allele were excluded from the AS and not-AS analysis (Supplementary Data 1).

*Normalization of read counts.* To account for sequencing and composition biases in the scRNA-seq data, gene count normalization was performed using the *scran* (v.1.12.1) R package[66]. Cell clusters were defined independently for each time point through the *computeSumFactors* function and the clusters parameter. Cluster-based scaling factors were then deconvoluted into cell-specific factors (Supplementary Data 2). The cell-specific scaling factors were derived from the autosomal not-AS count matrix and were used to compute the normalized CPM values for the $g$th gene in the $c$th cell of both the AS and not-AS count matrices, given by

$$\text{CPM}_{g,c} = 10^6 \cdot \frac{e_{g,c}}{f_c \cdot \sum\limits_{g} e_{g,c}}, \tag{2}$$

where $f_c$ is the scaling factor for cell $c$ and $e_{g,c}$ is the number of UMI counts overlapping gene exons for any gene $g$ in cell $c$. AS counts ($e_{g,c}^{B6}$ and $e_{g,c}^{Cast}$) were normalized in an analogous manner.

Bulk RNA-seq was normalized in a similar manner with sample-specific scaling factors ($f_c$) being computed with the TMM method[67] based on the not-AS autosomal count matrix, using the *calcNormFactors* function of the *edgeR* (v.3.26.8) R package.

*X:A ratio.* To account for the larger number of autosomal genes compared to X-linked genes (8880 and 374 genes, respectively) we used a bootstrapping procedure adapted from a previous study[32]. For each cell we randomly sampled with replacement a set of autosomal genes of the same size of the X-linked gene set (size = 374) and computed the ratio between the average X-linked expression and the average expression across the sampled autosomal genes. This step was repeated 1000 times, and the X:A ratio for each cell was estimated as the median across all 1000 bootstrap ratios. Only genes retained through the gene filtering step (see "Gene filtering" above) were included in the analysis.

*Pseudotime analysis.* Pseudotime trajectories allow to order single cells according to a biological process of interest and to identify key paths or branches corresponding to alternative cellular states. We analyzed single cell trajectories using the Monocle2 algorithm (*monocle* R package v.2.12.0)[24,68,69]. Starting from high dimensional data, this method projects cells into a lower dimensional space by constructing a principal graph and ordering them according to a pseudotime trajectory. Similarly to Cacchiarelli et al.[70], a set of ordering genes was defined as the 500 most DEGs over time, identified using the *differentialGeneTest* R function. The DDRTree method was used to project the cells into a two-dimensional space based on the expression of the selected genes, and simultaneously learn a graph structure into this space[69]. Pseudotime values were then estimated as the distance of each cell from the root of the graph, which is defined as the state with the highest number of cells sampled at day 0 of differentiation. Finally, the cell-wise scaled pseudotime was computed dividing the estimated values by the maximum across all cells (Supplementary Data 2).

*Dimensionality reduction (UMAP).* After log-transformation of the normalized not-AS gene expression values (Eq. 2), the most variable features were defined as the 500 genes with the highest variances across all cells. The number of features was further reduced through a principal component analysis (PCA) based on the centered expression levels of the selected genes using the *pca* R package (v.1.76.0). The top-50 principal components (PCs) were provided as input to the UMAP dimensionality reduction method[25] to further reduce dimensionality and visualize cells in a two-dimensional space, using the *umap* R package (v.0.2.3.1).

*RNA velocity analyses.* For RNA velocity analysis, read counts mapping completely to exonic regions, termed "spliced" (S), and those that overlapped with intronic regions, termed "unspliced" (U) were used.

*Global RNA velocity analysis.* We performed RNA-velocity analyses based on the not-AS spliced and unspliced count matrices, removing genes with low average spliced (≤1 counts) or unspliced (≤0.5 counts) expression across all cells[26]. RNA velocities were then calculated using the *gene.relative.velocity.estimates* function from the *velocyto.R* (v.0.6) R package with setting the cell neighborhood size to $kCells = 20$, performing a fit on the top and bottom 2.5% quantiles (*fit.quantile* = 0.025), and setting the remaining parameters to their default values. The estimated velocities were then projected onto the UMAP embedding and locally summarized through a vector field representation of single cell velocities.

*X-chromosomal RNA velocity.* To analyze XCI progression by means of the RNA velocity concept, we used the estimated RNA velocities for X-linked genes to predict the future transcriptional state of the X chromosome in each cell. The predicted expression of X-linked genes was then projected on a lower dimensional space. For this, we computed the fraction $l_{g,c}$ of spliced UMI counts assigned to the B6 allele (B6-fraction) for each gene $g$ on chromosome X in cell $c$, given by

$$l_{g,c} = \frac{s_{g,c}^{B6}}{s_{g,c}^{B6} + s_{g,c}^{Cast}} \in [0, 1], \tag{3}$$

where $g$ represents any gene on chromosome X for which the RNA-velocity model could be fitted, and $s_{g,c}^{B6}$, $s_{g,c}^{Cast}$ are the numbers of AS spliced reads of gene $g$ in cell $c$ on the B6 and Cast alleles, respectively. We then applied principal component analysis on the $\mathbf{L} = [l_{g,c}]$ matrix of the B6-ratios. The loadings of the first two principal components correspond to those X-linked genes which account for most of the variance in the data and therefore are the most important in determining differential silencing between the two alleles across cells. We then projected, for each cell, the vectors of predicted spliced X-linked transcripts obtained from the RNA velocity analysis on the space defined by the first two principal components, and visualized the average vector for each cell in a vector field which ultimately corresponds to different silencing trajectories.

*Definition of the predicted X-chromosomal change ΔX.* To compare gene expression profiles between cells that have initiated XCI and those that have not yet started to silence the X chromosome, we used the RNA velocity fit to identify cells that have initiated XCI. We computed the predicted change in X-chromosomal gene expression $\Delta X_c$ for each cell $c$ as

$$\Delta X_c = \frac{\sum\limits_{g} M_{g,c}}{\sum\limits_{g} P_{g,c}}, \tag{4}$$

where $g$ represents any gene on chromosome X for which the RNA velocity model could be fitted, and $M_{g,c}$, $P_{g,c}$ represent the measured and predicted normalized expression states for gene $g$ in cell $c$, respectively. The measured expression is computed from the exonic (spliced) read counts, while the predicted expression is derived by the RNA-velocity fit between the normalized spliced and unspliced expression (estimated with the *velocyto.R* (v.0.6) R package).

*Differential expression analyses.* In order to identify genes which were differentially expressed between cells expressing Xist at a high level versus cells expressing Xist at a low level, we clustered the cells with $K$-means clustering ($K = 7$, *kmeans* function from the *stats* R package) based on the log-transformed normalized Xist expression values $\log_{10}(\text{CPM}_{\text{Xist},c} + 1)$, where $\text{CPM}_{\text{Xist},c}$ are the normalized counts of the Xist gene in the $c$th cell. Cells belonging to the top three $K$-means groups, i.e., high expression of Xist, were classified as Xist-high, while cells belonging to the bottom 3 $K$-means groups, i.e., low expression of Xist were classified as Xist-low (Supplementary Data 2). $K$ was set in a way to minimize the within-cluster sum of squares value, while ensuring a minimum number of 50 cells in the Xist-high and Xist-low groups at each time point of differentiation. The set of DEG between Xist-high and Xist-low cells were identified at each time point through the MAST differential expression analysis method, using the *zlm* function from the *MAST* (v.1.10.0) R package[39]. In brief, a two-part generalized Hurdle model was fitted to the normalized CPM expression values. We set dummy variables to represent the two cell groups and the proportion of detected genes (UMI count > 0) per cell as model predictors. The significance between the two groups of cells was then assessed by a $\chi^2$ likelihood ratio test. The genes with a Benjamini–Hochberg adjusted $p$ value smaller or equal to 0.05 were deemed as significantly differentially expressed between the two cell groups[71]. Notably, the MAST method was only applied to compare groups containing at least ten cells. For this reason, the cells at day 0 were excluded from the analysis.

In order to identify differentially expressed genes between cells with high and low change in X-chromosomal gene activity, as quantified by $\Delta X_c$ (as defined in Eq. 4), we clustered cells with $K$-means clustering ($K = 3$, *kmeans* function from the *stats* R package) separately for each time point based on $\log_2(\Delta X_c)$. At each time point the set of DEG between the cells in the top and bottom clusters were identified through the MAST differential expression analysis method, as described above. The results are provided in Supplementary Data 3.

*Differential silencing analysis. XCI progress definition.* In order to compare gene-specific silencing dynamics between the two alleles, we estimated AS silencing half-times and compared them between the B6 and Cast X chromosomes. To account for the overall faster silencing on the Cast allele, we estimated the silencing half-time for each gene relative to the overall silencing state of the rest of the chromosome. To quantify the silencing state, we computed a measure we termed XCI progress (XP), which can be interpreted as the percentage of X-chromosomal gene activity that has already been silenced. It was computed for each cell $c$ that expressed Xist monoallelically as

$$\text{XP}_c = 100 \cdot \left( 1 - \frac{\sum\limits_{g} e_{g,c}^{\text{Xi}} + 0.01}{\sum\limits_{g} e_{g,c}^{\text{Xa}} + 0.01} \right), \tag{5}$$

where $g$ represents any gene on chromosome X except Xist, Xi refers to the Xist-expressing chromosome, Xa to the Xist-negative allele (Supplementary Data 2) and $e_{g,c}^{\text{Xi}}$, $e_{g,c}^{\text{Xa}}$ are the numbers of AS molecules overlapping exons of gene $g$ in cell $c$ on the Xi and Xa alleles, respectively. For cells where $\text{XP}_c$ was negative, it was set to 0. $\text{XP}_c$ is a proxy for the extent of inactivation that has already occurred on Xi in cell $c$. Intuitively, a high $\text{XP}_c$ value indicates that the cell has already silenced a substantial number of X-linked genes, while a value proximal to zero indicates that the two alleles have similar gene expression levels.

*Cell binning.* Since allelic expression of individual genes in single cells tends to be noisy, we grouped cells with comparable X-inactivation status and quantified overall silencing (XP) and gene-specific silencing by aggregating read counts of all cells in each group. The following steps were performed separately for cells silencing the B6 (Xist MA-B6) and Cast (Xist MA-Cast) X chromosome, respectively.

First, cells that had not yet initiated XCI with $\text{XP}_c \leq 10\%$ were excluded from the analysis. The XP range covered by the remaining cells was divided into ten equally sized bins. The overall XCI progress for each bin $b$, denoted as $\text{XP}_b$ was then calculated by aggregating the total number of AS molecules across all cells belonging to that bin as follows

$$\text{XP}_b = 100 \cdot \left( 1 - \frac{\sum\limits_{c \in b} \sum\limits_{g} e_{g,c}^{\text{Xi}} + 0.01}{\sum\limits_{c \in b} \sum\limits_{g} e_{g,c}^{\text{Xa}} + 0.01} \right), \tag{6}$$

where $XP_b$ quantifies the extent of X inactivation across all cells assigned to a bin, and $e_{g,c}^{Xi}$, $e_{g,c}^{Xa}$ denote the AS read counts from the inactive and active allele, respectively, for a gene $g$ on chromosome X (except Xist) in cell $c$. Next, the extent of silencing of each X-linked gene was quantified in each bin as the Xi-to-Xa expression ratio $r_{g,b}$ of that gene, again by aggregating the total number of AS molecules

$$r_{g,b} = \frac{\left(\sum_{c \in b} e_{g,c}^{Xi}\right) + 0.01}{\left(\sum_{c \in b} e_{g,c}^{Xa}\right) + 0.01}, \tag{7}$$

where $r_{g,b}$ is a proxy for the extent of inactivation of a specific gene in bin $b$. Intuitively, a value of $r_{g,b}$ close to zero indicates that the $g$th X-linked gene has been completely silenced on the Xi allele, while a value proximal to one indicates that the two alleles have similar gene expression levels. For each gene and allele, the analysis was restricted to the bins containing a minimum of 5 cells and at least 25 AS counts, and to genes with a minimum of 5 such bins. The inactivation ratio for each gene in Eq. 7 needs to be corrected for basal expression skewing due to genetic variation between the two alleles. Basal skewing was estimated from the B6-to-Cast ratio in Xist-negative cells at day 0 ($c'$) as

$$s_g = \frac{\left(\sum_{c'} e_{g,c'}^{B6}\right) + 0.01}{\left(\sum_{c'} e_{g,c'}^{Cast}\right) + 0.01}, \tag{8}$$

where $c'$ refers to Xist-negative cells at day 0 with $0.4 \le R_c \le 0.6$, as defined in Eq. 1. The Xi-to-Xa ratio was then normalized to the baseline ratio in the absence of XCI as

$$r_{g,b}^* = \frac{r_{g,b}}{r_{g,0}}; \text{ with } r_{g,0} = \begin{cases} s_g, & \text{for Xi = B6} \\ 1/s_g, & \text{for Xi = Cast} \end{cases} \tag{9}$$

*Silencing halftimes.* To quantify gene- and AS gene silencing rates, we modeled the normalized Xi-to-Xa ratio $r_{g,b}^*$ for each gene $g$ on chromosome X using an exponential decay function of the XCI progress XP

$$E[\log_2(r_{g,b}^*)] = -\beta_g \cdot XP_b, \tag{10}$$

where $\beta_g$ denotes the relative silencing rate. Eq. 10 was fitted to the binned and normalized Xi-to-Xa ratios $r_{g,b}^*$, separately for cells silencing the B6 and Cast chromosome, to estimate the AS silencing rates $\beta_g^{B6}$ and $\beta_g^{Cast}$. The silencing halftimes ($XP_{50,g}$) were then computed as

$$XP_{50,g} = \frac{1}{\beta_g} \tag{11}$$

The $XP_{50,g}$ values greater than 100 were set equal to 100. To classify genes according to their AS silencing dynamics (fast, intermediate, slow, escape), $K$-means clustering ($K = 4$) was performed on the $XP_{50,g}$ values of each allele separately (Supplementary Data 2).

To test whether silencing rates were significantly different between the two alleles, both alleles were fitted simultaneously with the equation

$$E[\log_2(r_{g,b}^*)] = \beta_{1,g} \cdot XP_b + \beta_{2,g} \cdot XP_b \cdot a; \text{ with } a = \begin{cases} 0 & \text{for Xi = B6} \\ 1 & \text{for Xi = Cast} \end{cases}, \tag{12}$$

where $\beta_{1,g}$ and $\beta_{1,g} + \beta_{2,g}$ estimate the silencing rates on the B6 and Cast allele, respectively, for the X-linked gene $g$. An ANOVA $F$ test was then used to assess whether the parameter $\beta_{2,g}$ was significantly different from 0 (H$_0$: $\beta_{2,g} = 0$). Any gene with a Benjamini–Hochberg adjusted $p$ value smaller or equal to 0.05 was deemed as differentially silenced between the two alleles.

*Bulk RNA-Sequencing (TXΔXic) data analysis.* To validate differences in gene silencing rates on the two alleles identified from the scRNA-seq data, silencing dynamics were analyzed in bulk RNA-seq data in cell lines, where the Xic is deleted either on the Cast (TXΔXic_{Cast}) or on the B6 (TXΔXic_{B6}) allele. To compare chromosome-wide silencing kinetics of the two alleles, we computed the Xi-to-Xa ratios for all X-linked genes by summing up counts across replicates as

$$r_g = \frac{\sum_s e_{g,s}^{Xi} + 1}{\sum_s e_{g,s}^{Xa} + 1}$$

$$\text{with } \begin{cases} Xi = B6 \& Xa = Cast, & \text{if } s \in TX\Delta Xic_{Cast} \\ Xi = Cast \& Xa = B6, & \text{if } s \in TX\Delta Xic_{B6}, \end{cases} \tag{13}$$

where $e_{g,s}^{B6}$ and $e_{g,s}^{Cast}$ represent the AS counts of the B6 and Cast alleles respectively for all X-linked genes $g$ in replicate sample $s$. Genes within the deleted region (chrX: 103,182,257–103,955,531, mm10) and genes with less than 50 AS counts were excluded from the analysis. These allelic ratios were compared between the two deletion lines at each time point through a paired two-sided Student's T-test statistic.

To validate differentially silenced genes identified through scRNA-seq, the Xi-to-Xa ratio was computed for each gene $g'$ (Klhl13, Pir, Hprt) as

$$r_{g',s} = \begin{cases} e_{g',s}^{B6}/e_{g',s}^{Cast}, & \text{if } s \in TX\Delta Xic_{Cast} \\ e_{g',s}^{Cast}/e_{g',s}^{B6}, & \text{if } s \in TX\Delta Xic_{B6} \end{cases} \tag{14}$$

To account for genetic skewing, the allelic ratios were normalized to the respective ratio at day 0 before onset of XCI, averaged across replicates, analogous to the procedure described for scRNA-seq data in the previous section.

*Pyrosequencing data analysis.* To assess gene silencing we computed the Xi-to-Xa ratio $r_{g,s}$ based on pyrosequencing data in the two ΔXic deletion lines as

$$r_{g,s} = \begin{cases} p_{g,s}/(1 - p_{g,s}), & \text{if } s \in TX\Delta Xic_{Cast} \\ (1 - p_{g,s})/p_{g,s}, & \text{if } s \in TX\Delta Xic_{B6} \end{cases} \tag{15}$$

where $p_{g,s}$ represents the percentage of B6-molecules observed for the gene $g$ (Klhl13, Pir, Hprt, Rlim, Atrx, Renbp, Cul4b, and Prdx4) in replicate sample $s$. Aiming to account for baseline allelic detection skewing, the above ratios were normalized to the respective ratio at day 0 before onset of XCI, averaged across replicates, analogous to the procedure described for scRNA-seq data above.

For each gene deemed as differentially silenced between the two alleles (Klhl13, Pir, and Hprt), the normalized ratios were compared between the two deletion lines through an unpaired two-sided Student's $T$-test statistic. The average normalized ratios of 5 not differentially silenced genes (Rlim, Renbp, Cul4b, Prdx4, and Atrx) were averaged across replicates and compared between the two deletion lines through a Wilcoxon signed-rank test.

**Reporting summary**. Further information on research design is available in the Nature Research Reporting Summary linked to this article.

## Data availability

ScRNA-seq and bulk RNA-seq data generated during this study are available via GEO with identifier GSE151009. All other relevant data supporting the key findings of this study are available within the article and its Supplementary Information files or from the corresponding author upon reasonable request. Source data are provided with this paper.

## Code availability

All code used in this paper is available on https://github.com/EddaSchulz/Pacini_paper [https://doi.org/10.5281/zenodo.4647585][72].

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

## Acknowledgements

We would like to thank Maud Borensztein and Marc Friedlander for critical feedback on the manuscript; Thorsten Mielke and Beatrix Fauler for imaging support; Sven Klages for initial data processing and the IT service at the MPI for Molecular Genetics for

maintenance and support of a computing cluster. This work was supported by the Max Planck Research Group Leader program, E:bio Module III-Xnet grant (BMBF 031L0072) and Human Frontiers Science Program (CDA-00064/2018) to E.G.S. G.P. was part of the IMPRS for Biology and Computing.

## Author contributions

E.G.S., A.M., and G.P. conceived the present work. I.D. and N.M. performed the experiment with the input from E.S. and B.T. G.P. performed all the analyses and V.M. generated the ΔXic cell lines. E.G.S. and A.M. wrote the paper with the contribution of G.P.

## Funding

## Competing interests

The authors declare no competing interests.
