## [Peer Review File · Nature Communications]

REVIEWER COMMENTS

Reviewer #1 (Remarks to the Author):

Using allele-specific single-cell RNA-seq in differentiating mouse embryonic stem cells, the authors perform an integrated analysis of cell differentiation, upregulation of RNA Xist, and gene silencing during early development to assess the onset of random inactivation of one of the two X chromosomes in XCI. Importantly, the authors validated some of their experimental/computational findings using “an orthogonal experimental approach”, which I found it to be a strong aspect of the paper. Due to my background in computational biology, my remarks below are focused on the computational aspects of the paper. In general, I followed the paper with interest, and I think that it is worth publication, provided that the biological developments are novel. However, I found two main issues that need attention: (i) many analysis steps require specification of thresholds which are taken arbitrarily and with no explanation why the chosen values are appropriate, and (ii) the notation and description of the computational methods used in this paper are quite difficult to follow due to complex and inconsistent notation and explanations. The second issue is easy to fix if the authors make a diligent effort to revise the methods section. However, the first issue seems to me more crucial for the results and discussions provided in the paper, since it seems to me that these depend on the authors’ choices for these thresholds, which are many. You should provide some discussion and insights about this issue.

Due to formatting issues, I submit my entire review as a separate PDF file.

(Please see page 5 of the Peer Review File for attachment)

Reviewer #2 (Remarks to the Author):

This is an interesting and well performed study of X chromosome inactivation in a mouse pluripotent stem cell system with allelic resolution. A number of novel features of XCI are revealed, including the high-resolution temporal dynamics of XCI with single-cell resolution, the identification of putative Xist regulators and differences between genetic background. The manuscript is well written.

I have some questions which upon reading remained open for me:

1. X chromosome upregulation. Can you also refer to the Lentini/Reinus BioRxiv 2020 work and indicate where your findings corroborate or differ from this other study? The second paragraph of page 6 may be a good place to further discuss this.
2. Why is the upregulation of Xa not shown in Figure 1A?
3. Figure 1. Can you show a UMAP and project the X/A ratio on it? One plot for Xist+ cells and one for Xist- cells?
4. Figure 4E: difficult to read because it is not clear if the legend belongs to the figure or not and it takes a while to figure out.
5. Page 7, penultimate line: replace 'reveal' by 'confirms'
6. Page 11: last paragraph -> We then integrated 'X' chromosome-wide.... Specificity it is the X chromosome.
7. Page 13, third sentence before last "XCI and differentiation appeared to be partially independent process". It was not clear to me how the figure shows that.
8. Can you calculate allele-specific X/A ratios (XCast/A and XMus/A ratios) and plot that along the pseudotime for cells that inactivate the Mus or the Cast allele (2 plots)? This could go into Figure 5 or 1.
9. Profiling random XCI in more physiological contexts, including primary human cells in vivo. How will the experimental and computational strategies developed here help? What if there are much less SNPs

in humans and these are not always known? Can you also mention the expected limitations of the human system? You may also want to refer to Tukiainen/MacArthur Nature 2017.

10. Page 24. First paragraph 'genome genome'.

11. Page 26. X:A Ratios. Were non-expressed genes removed from the analysis of X:A ratios?

12. Figure S1C. Remove extra dotted line on the right.

Reviewer #3 (Remarks to the Author):

Reviewer report for "Integrated analysis of Xist upregulation and gene silencing at the onset of random X-chromosome inactivation at high temporal and allelic resolution" by Pacini et al.

SUMMARY:

In this study Pacini et al aimed to answer questions related to the establishment of random X-chromosome inactivation (XCI), more specifically how Xist expression features (level, allelic transcription, and speed of upregulation) affect silencing and what factors trigger Xist expression.

The authors used mouse embryonic stem cells (mESCs) derived from F1 hybrid embryos (Castaneous x C57/Bl6; providing allelic resolution of gene expression for SNP-containing transcripts) that were differentiated in vitro to induce XCI. Individual cells were captured throughout differentiation (Day: 0, 1, 2, 3, 4) using a C1 Fluidigm and were subjected to single-cell RNA-sequencing (scRNAseq) by a 3' strand-sensitive method (C1-HT). Bioinformatical analyses were used to identify allelic expression patterns that correlated with Xist expression features. Finally, the authors used CRISPR/Cas9-mediated genome editing on mESCs to generate lines with deletions of the X-chromosome inactivation center (Xic), including the Xist gene and other genes in Xic (deletion of ~800k region), at either the CAST or Bl6 allele. This system of predictably skewed XCI was used to perform bulk analysis of selected genes using pyrosequencing.

The manuscript well written and experiments as well as analyses are intuitive and sound. The generated dataset is valuable for the community studying XCI and other allelic events during mESC differentiation. Overall, I find the study valuable and that the manuscript provides novel findings, and I am therefore positive to seeing it becoming published. There is however room for improvements by reinforcing some of the findings. I also find some conclusions a bit overreached and that interpretations based on observed correlations should be moderated. Below you will find points that I think should be addressed.

MAJOR POINTS:

[1] The authors should be more explicit to the reader about the benefits as well as the drawbacks of using a 3' method for the allelic scRNAseq analyses. This distinction becomes particularly important as similar allelic scRNAseq studies of in vitro and in vivo differentiation using the same F1 cross have been performed by others but using full-length-coverage smart-seq2 (full-length and high sensitivity; G. Chen Genome Research 2016, S. Cheng Cell Reports 2019) or smart-seq3 (full-length, high sensitivity, 5' strand specificity; A. Lentini bioRxiv 2020). In the context of the paper, I find strand-specificity and relatively good Xist detection to be the main benefits of the used 3' C1-HT method, while the drawback being lower sensitivity to detect genes as well as much lower number of genes detected at allelic resolution (due to lack of SNPs in read-covered 3' regions). Importantly, strand specificity allows a direct distinction between Xist and antisense Tsix transcripts, which is not possible for smart-seq2 data (although transcripts can be bioinformatically distinguished based on known exon/intron structures and junctions). Also, Xist is known not be detected inefficiently by smart-seq2, likely due to full-length amplification of the method together with Xist transcript's extreme length as well as possible inaccessibility of Xist RNA for reverse transcription (being a chromatin-bound lncRNA). Judging from the data, Xist seems to be better detected by the 3' C1-HT method. If the authors agree on these points, it would strengthen the paper to better highlight the mentioned strong points of the 3' C1-HT and that this distinguishes the newly generated scRNAseq data from that of previous studies. Equally important is to be transparent with the method's drawback (mainly lower sensitivity and lower number of genes with allelic resolution) to the read. This is easily fixed by rewording a few sentences, but I marked this

as a major point because many readers may lack sufficient knowledge in details of scRNAseq methodology to appreciate these issues.

[2] It would be valuable to get an idea about Xist cloud (or speck) formation especially for cells/stages in which Xist is biallelically transcribed, as such cells were shown to have biallelic reduction of X-gene expression. Showing strand-specific Xist RNA-FISH data at the stage of highest biallelic Xist frequency would strengthen the paper. However, if the authors are convinced that RNA-FISH data shown in previous studies performed by themselves or others sufficiently covers this and that limited value would be gained by new RNA-FISH data, the authors can instead more clearly describe previous findings of biallelic Xist cloud formation in the text.

[3] Nanog as the initial trigger of Xist upregulation: Using two different approaches, the authors propose downregulation of Nanog as the earliest event in XCI. However, I think the authors are over-extrapolating the data here as I do not see how they can pinpoint Nanog as the “initial” trigger of Xist upregulation based on the observed correlations. How about all other potential factors that might have been missed (not detected)? I also think that usage of terminology such as “initial” trigger is both risky and unnecessary, because unknown upstream as well as intermediate factors might exist. Please provide full evidence that Nanog is truly the initial trigger or moderate your conclusions and statements according to the evidence. Additional insights might be provided by RNA-seq using the TXdelta cell lines described in the last section of the study.

MINOR POINTS

[4] That XCI is partially initiated (reduced gene expression) on both alleles in cell expressing Xist biallelically is clearly a main finding of the study. This finding should be highlighted in the abstract.

[5] Implementation of computational methodology to analyze XCI at the single-cell level: The use of RNA velocity to look at allelic silencing is one of the most exciting but is only used for visualization. I think it could strengthen the manuscript if the output from this analysis section was utilized more in the manuscript.

[6] It would be nice to see pseudotime results incorporated with the analysis to show how the allelic expression of X-genes behave during the differentiation process rather than relative to Xist. Differential expression could also be performed along parts of the pseudotime trajectory using MAST to identify early events in a manner that is less dependent on data timepoints as XCI is variable.

[7]. Estimation of silencing dynamics. The authors estimate the silencing rate of individual genes for each allele along the XCI process and identify both gene- and strain-specific effects which is supported by reanalysis of published data as well as their experimental cell line model. Here, it would be interesting to see how the silencing rate behaves across the X chromosome as previous papers have suggested that silencing occurs more rapidly in close vicinity to Xist whereas this may not be true at the single-cell level. Does the 3' data have enough gene coverage with allele-resolved expression to perform this analysis?

[8] Biallelic Xist expression together with decreased expression of both alleles have been reported in female human preimplantation development as well as in human ESCs, i.e. so called “dampening” which would be in line with partial XCI on both X-alleles [S. Petropoulos Cell 2016; A Sahakyan Cell Stem Cell 2017]. In the context of main results of the current study, i.e. partial XCI and reduced expression from both alleles in female mouse cells with biallelic Xist expression, it would be interesting to read a discussion paragraph on potential translation between the observations in human and mouse.

[9] I appreciate the Methods section describing the computational calculations in detail, but I did not find a link to the computational code. Please make the computational code available.

[10] Fig. 1F: The Xist- trend here is quite interesting. Would be good to know if these cells also downregulate Nanog and if this trend is chromosome-wide or driven by few genes.

[11] p.6 “Around 4% of all reads could be mapped in an allele-specific manner (Fig. S1A).” It should also be stated how many genes per cell were detected at the allelic resolution (i.e. having coverage over SNP positions).

[12] p.8. "We used two different approaches to identify such candidate regulators, based on differential expression and correlation analysis, respectively, to ensure robustness of the results (Supplemental Table S3)." These are both indirect approaches. An alternative approach would be to stratify expression over pseudotime to find genes that react prior to Xist. The authors could consider to use MAST for this after excluding day0 cells and genes with the highest negative FC would be possible candidates to check.

[13] p.20 More Xist from B6, can you really say this? Is this not confounded by potential preferential detection (binding, accessibility) between the strains? How did it look in your bulk data?

FINAL REMARKS: I hope these comments will help the authors to further improve the manuscript. I find the study and data very valuable.

Reviewer #1 Attachment

Using allele-specific single-cell RNA-seq in differentiating mouse embryonic stem cells, the authors perform an integrated analysis of cell differentiation, upregulation of RNA Xist, and gene silencing during early development to assess the onset of random inactivation of one of the two X chromosomes in XCI. Importantly, the authors validated some of their experimental/computational findings using “an orthogonal experimental approach”, which I found it to be a strong aspect of the paper. Due to my background in computational biology, my remarks below are focused on the computational aspects of the paper. In general, I followed the paper with interest, and I think that it is worth publication, provided that the biological developments are novel. However, I found two main issues that need attention: (i) many analysis steps require specification of thresholds which are taken arbitrarily and with no explanation why the chosen values are appropriate, and (ii) the notation and description of the computational methods used in this paper are quite difficult to follow due to complex and inconsistent notation and explanations. The second issue is easy to fix if the authors make a diligent effort to revise the methods section. However, the first issue seems to me more crucial for the results and discussions provided in the paper, since it seems to me that these depend on the authors’ choices for these thresholds, which are many. You should provide some discussion and insights about this issue.

1. Abstract. Cell differentiation, Xist upregulation and <gene> silencing ...
2. Abstract (and elsewhere). You are saying that the analysis is performed at “high temporal resolution”. Is this really “high”. You are only using five time points that are one day apart. Is this considered “high” for this application?
3. Introduction (and elsewhere). You are using the words “stochastic” and “random” interchangeably. You may want to stick to “random” for uniformity and to avoid possible confusion.
4. Introduction. Is the developed framework analytical? It seems experimental/computational to me.
5. I will suggest that you report *P*-values using the following convention suggested by NEJM: “In general, *P*-values larger than 0.01 should be reported to two decimal places, and those between 0.01 and 0.001 to three decimal places; *P*-values smaller than 0.001 should be reported as $p < 0.001$. Notable exceptions to this policy include *P*-values arising from tests associated with stopping rules in clinical trials or from genome-wide association studies.” Differences in *P*-values below 0.001 are insignificant and provide the same information. *P*-values of 2.2×10^{-16} and 6.2×10^{-14} are both zero for all practical purposes (see page 6).
6. Page 5. It will be better to say “Pseudotime analysis with Monocle and dimensionality reduction using UMAP revealed ...”, since Monocle is applied first and then you use UMAP for visualization.
7. Page 6. Your classification of cells as Xist-positive uses a threshold of 5 UMI counts. How did you choose this threshold, which seems to me arbitrary? Will the choice affect your conclusions? A similar remark applies for the two classification schemes at the bottom of page 6, as well as for other thresholdings you used in the paper (e.g., page 25).
8. Page 8. Your *K*-means clustering defines cells as Xist-low. However, the term Xist-low was also used on page 6 in a different classification scheme. This is confusing.
9. Pages 8 & 11. A few things are repeated on these pages. Please consolidate them.
10. Pages 8-9. Reference Sup. Table S3 and write everywhere in the paper Benjamini-Hochberg adjusted *P*-value or use the terminology of *Q*-value to indicate FDR control, which is more preferable. Why

are you using a correlation threshold of 0.25? Again, it seems arbitrary to me and can affect results and conclusions.

11. Page 12. You discuss the onset of gene silencing in Fig. 4E and you provide the P -values at days 2 and 3. I think you should display all P -values in this figure, as you did for Fig. 2C. This may explain how you decided that there is onset in day 2 (I take it that the P -values in days 0 and 1 are > 0.05).
12. Figure 6(B-D). Indicate in the caption what the displayed P -values mean.
13. Page 24. I think you should not place definitions of various quantities here. It makes it very difficult for understanding subsequent formulas. I think you should include definitions of various quantities used by a formula, after the formula is displayed. This will help the reader understand what the formula is about without having to recall the meaning of different terms. For example, after Eq. 1 you should write “where g denotes a gene in chromosome X , $c = \{1, 2, \dots, C\}$ denotes a particular cell, ...” Similarly, for Eqs. 2-9. You can simply use R_c instead of XR_c .
14. Page 24. You can use s_c instead of sf_c . You must simplify your notation throughout.
15. Page 26. What are the Y 's used in Eq. 2? I do not understand this formula. Does it mean that the formula is used for every Y in the parenthesis? If that is so, then CPM will depend on the particular Y used. Place 10^6 on the left-hand side as $10^6 \times Y_{g,c} / (sf_c \times \sum_{g' \in chrX} Y_{g',c})$. You should be using different g 's otherwise it is confusing. Why do you multiply this with 10^6 ?
16. Page 26. Write $CPM_{g,c}$ or something simpler. In addition, using matrix X for the count matrix is confusing because you use X for X^{B6} and X^{Cast} . Perhaps you should use E^{B6} and E^{Cast} , since they are related to exon-overlapping, and \mathbf{M} for the count matrix (usually matrices in math are denoted by bold, non-italic symbols).
17. Page 26. You refer to K -means clustering ($K = 7$) but before you referred to k -means clustering with $k = xx$. Be consistent. Write also “Genes with a Benjamini-Hochberg derived Q -values ≤ 0.05 were deemed ...”
18. Page 26. You must explain the bootstrapping procedure you are using here and the purpose of bootstrapping. Although it seems that this is explained in the provided reference, the reader should not have to go to this reference to find what is going on here.
19. Page 27, RNA velocity analysis. Now you are using S and U as count matrices. I think you should use bold non-italic again and write $\mathbf{S} = [S_{g,c}]$ and the same for U . Also, replace Λ in a formula with “and”, and do similarly on page 28.
20. Page 28. Denote L and D as matrices (bold, non-italic). What is a “B6-ratio vector”? Why do the dots mean in $l_{.,i}$ and $l_{.,j}$ in the correlation term on the right-hand-side of Eq. 4? Why did you suddenly change from denoting a cell by c to now denoting it by i or j ?
21. Page 28. In Eq. 5 write XP_c and not XP_c , or something simpler, and place 100 on the left of the parenthesis as $100 \times (\dots)$ or better as $100 \times \max \{0, 1 - \sum \dots / \sum \dots\}$. Define terms.
22. Page 29. Why using a threshold of 10% and not any other threshold, say 5% or 20%? Before you used XP_c and now you are using XP_b . You should not be doing that, it is confusing and mathematically wrong. Same issue in $X_{b,g}^{X_i}$ and $X_{b,g}^{X_a}$. Define these terms.

23. Pages 29-30: Define terms in Eq. 6. The description after Eq. 6 and to the end of Methods on page 30 is difficult to follow due to complex and difficult to read notation. It must be revised. You really need to try to make the math readable and understandable.

Reviewer #1

Using allele-specific single-cell RNA-seq in differentiating mouse embryonic stem cells, the authors perform an integrated analysis of cell differentiation, upregulation of RNA Xist, and gene silencing during early development to assess the onset of random inactivation of one of the two X chromosomes in XCI. Importantly, the authors validated some of their experimental/computational findings using “an orthogonal experimental approach”, which I found it to be a strong aspect of the paper. Due to my background in computational biology, my remarks below are focused on the computational aspects of the paper. In general, I followed the paper with interest, and I think that it is worth publication, provided that the biological developments are novel. However, I found two main issues that need attention: (i) many analysis steps require specification of thresholds which are taken arbitrarily and with no explanation why the chosen values are appropriate, and (ii) the notation and description of the computational methods used in this paper are quite difficult to follow due to complex and inconsistent notation and explanations. The second issue is easy to fix if the authors make a diligent effort to revise the methods section. However, the first issue seems to me more crucial for the results and discussions provided in the paper, since it seems to me that these depend on the authors’ choices for these thresholds, which are many. You should provide some discussion and insights about this issue.

We want to thank the reviewer for these very useful comments that helped us to improve the manuscript significantly. We have revised the computational methods section extensively and we have performed additional analyses to ensure parameter robustness for several analyses.

1. Abstract. Cell differentiation, Xist upregulation and <gene> silencing ...

We have modified the abstract as suggested to improve clarity.

2. Abstract (and elsewhere). You are saying that the analysis is performed at “high temporal resolution”. Is this really “high”. You are only using five time points that are one day apart. Is this considered “high” for this application?

As pointed out by the reviewer it is very subjective what to consider “high” temporal resolution. We have therefore removed this notion from the manuscript.

3. Introduction (and elsewhere). You are using the words “stochastic” and “random” interchangeably. You may want to stick to “random” for uniformity and to avoid possible confusion.

On p. 1 we have substituted “stochastic” by “random”. The term “stochastic” is now used only in the context of the “stochastic model of XCI”, which is a fixed term in the X inactivation field.

4. Introduction. Is the developed framework analytical? It seems experimental/computational to me.

We have substituted “analytical” by “computational” as suggested (p.2).

5. I will suggest that you report P-values using the following convention suggested by NEJM: In general, P-values larger than 0.01 should be reported to two decimal places, and those between 0.01 and 0.001 to three decimal places; P-values smaller than 0.001 should be reported as $p < 0.001$. Notable exceptions to this policy include P-values arising from tests associated with stopping rules in clinical trials or from genome-wide association studies.” Differences in P-values below 0.001 are insignificant and provide the same information. P-values of 2.2×10^{-16} and 6.2×10^{-14} are both zero for all practical purposes (see page 6).

We agree with the reviewer that differences between very low p-values are unimportant for data presentation. We have modified the manuscript as suggested.

6. Page 5. It will be better to say “Pseudotime analysis with Monocle and dimensionality reduction using UMAP revealed ...”, since Monocle is applied first and then you use UMAP for visualization.

We have modified the text as suggested by the reviewer.

7. Page 6. Your classification of cells as Xist-positive uses a threshold of 5 UMI counts. How did you choose this threshold, which seems to me arbitrary? Will the choice affect your conclusions? A similar remark applies for the two classification schemes at the bottom of page 6, as well as for other thresholdings you used in the paper (e.g., page 25).

We agree that it is important to ensure that the conclusions from our analyses are robust to the choice of such (somewhat arbitrary) threshold values. To this end, we have repeated the analysis in Fig. 1f for several other threshold values, all of which show a reduction of the X:A ratio in Xist+ compared to Xist- cells (Fig. S3a). Moreover, we now show the Xist pattern classification for different threshold values in Fig. S5a, all of which show transient biallelic expression at day 2. Finally, we confirmed parameter robustness of the differential silencing analysis in Fig. S9c, showing that the three differentially silenced genes identified (Klhl13, Pir, Hprt) are found over a wide range of threshold values.

8. Page 8. Your K-means clustering defines cells as Xist-low. However, the term Xist-low was also used on page 6 in a different classification scheme. This is confusing.

To prevent confusion regarding the Xist-low category, we have renamed it in the Xist pattern classification in Fig. 2a-b to “1-5 Xist counts”.

9. Pages 8 & 11. A few things are repeated on these pages. Please consolidate them.

We are not sure what this comment refers to. However, page 8 has been largely rewritten due to additional analyses that have been added.

10. Pages 8-9. Reference Sup. Table S3 and write everywhere in the paper Benjamini-Hochberg adjusted P -value or use the terminology of Q -value to indicate FDR control, which is more preferable. Why are you using a correlation threshold of 0.25? Again, it seems arbitrary to me and can affect results and conclusions.

We have changed the notation to Benjamini-Hochberg adjusted p -value, which is more commonly used in Bioinformatics than q -value. The correlation threshold of 0.25 is indeed an arbitrary choice and was removed. We now show 20 genes with the highest correlation coefficient in Fig. 4e+f and provide a full list of correlated genes in Supplemental Table S3. The conclusions now mostly discuss genes that were significantly correlated or differentially expressed in several analyses (new Fig. 5) and are therefore independent of which genes are displayed in Fig. 4.

Regarding suppl. Table S3 we have now pointed out in the table description that FDR represents the Benjamini-Hochberg adjusted p -value.

11. Page 12. You discuss the onset of gene silencing in Fig. 4E and you provide the P -values at days 2 and 3. I think you should display all P -values in this figure, as you did for Fig. 2C. This may explain how you decided that there is onset in day 2 (I take it that the P -values in days 0 and 1 are > 0.05).

We now report p -values for day 1,2 and 3 as suggested (p.10). For day 0 and 4 the comparison between mono-allelic and bi-allelic cells cannot be performed, because no bi-allelic cells are detected at those time points.

12. Figure 6(B-D). Indicate in the caption what the displayed P -values mean.

We have modified the legend of this figure (now Fig. 7) to explain the statistical comparisons in more detail.

13. Page 24. I think you should not place definitions of various quantities here. It makes it very difficult for understanding subsequent formulas. I think you should include definitions of various quantities used by a formula, after the formula is displayed. This will help the reader understand what the formula is about without having to recall the meaning of different terms. For example, after Eq. 1 you should write “where g denotes a gene in chromosome X , $c = \{1,2, \dots, C\}$ denotes a particular cell, ...” Similarly, for Eqs. 2-9. You can simply use R_c instead of XR_c .

We are happy to follow the reviewer suggestion. We have removed the definition of the various quantities on page 24 but made sure that every quantity or symbol in each formula is defined after the formula has been introduced, following the notation suggested by the reviewer. In equation 1, the quantity XR_c has been now replaced with R_c .

14. Page 24. You can use s_c instead of sf_c . You must simplify your notation throughout.

In equation two (p. 28) we have simplified the notation of the normalization factor from sf_c to f_c (to avoid confusion with the spliced counts, which are also denoted by s). We also have tried to simplify the mathematical notation throughout the manuscript.

15. Page 26. What are the Y 's used in Eq. 2? I do not understand this formula. Does it mean that the formula is used for every Y in the parenthesis? If that is so, then CPM will depend on the particular Y used. Place 10^6 on the left-hand side as $10^6 \times Y_{g,c} / (sf_c \times \sum_{g' \in \text{chr}X} Y_{g',c})$. You should be using different g 's otherwise it is confusing. Why do you multiply this with 10^6 ?

We apologize for the confusing notation and hope that this section is easier to follow in the revised version of the manuscript. We have removed the variable Y and now introduce the CPM normalization only for the exonic counts E with stating that it was calculated in an analogous manner for the allele-specific analyses. Moreover, we have modified the formula following the reviewer's suggestion. In particular we now only use g for indicating the gene whose expression level is being normalized.

We multiply the normalized gene expression level by 10^6 according to the definition of Count-per-million (CPM), where the multiplication per 10^6 transforms very small numbers (in the order of 10^6 as result of the normalization procedure) to numbers in the order of 10, which are easier to handle in mathematical operations.

16. Page 26. Write $CPM_{g,c}$ or something simpler. In addition, using matrix X for the count matrix is confusing because you use X for X^{B6} and X^{Cast} . Perhaps you should use E^{B6} and E^{Cast} , since they are related to exon-overlapping, and \mathbf{M} for the count matrix (usually matrices in math are denoted by bold, non-italic symbols).

We were happy to follow the reviewer's suggestion and denoted the matrices of exon-overlapping reads with E^{B6} and E^{Cast} instead of X , and converted all matrices to bold notation throughout the manuscript (e.g. page 30)

17. Page 26. You refer to K-means clustering ($K = 7$) but before you referred to k-means clustering with $k = x$. Be consistent. Write also "Genes with a Benjamini-Hochberg derived Q -values ≤ 0.05 were deemed ..."

We apologize for the confusing notation. We have changed that and now use the notation K to indicate the number of clusters resulting from the application of the K-means algorithm throughout the entire method section, as well as the result section.

18. Page 26. You must explain the bootstrapping procedure you are using here and the purpose of bootstrapping. Although it seems that this is explained in the provided reference, the reader should not have to go to this reference to find what is going on here.

We have now extended the description of the bootstrap procedure used to compute the X:A ratio (page 28).

19. Page 27, RNA velocity analysis. Now you are using S and U as count matrices. I think you should use bold non-italic again and write $S = [S_{g,c}]$ and the same for U . Also, replace Λ in a formula with “and”, and do similarly on page 28.

We have converted the symbols of matrices to bold and denoted them as suggested, albeit using small letters to indicate matrix elements. Moreover, we have rewritten the sentence that previously contained the symbol Λ .

20. Page 28. Denote L and D as matrices (bold, non-italic). What is a “B6-ratio vector”? Why do the dots mean in $l.,i$ and $l.,j$ in the correlation term on the right-hand-side of Eq. 4? Why did you suddenly change from denoting a cell by c to now denoting it by i or j ?

We have found a way to simplify the X-chromosomal RNA velocity analysis, which is reflected by a much simplified (and shortened) methods section, describing these analyses (p. 30). We simplified the notation and introduced only matrix L (in bold). We also added a definition for the B6-ratio (renamed to B6-fraction), defined, for each gene and for each cell, as the fraction of spliced UMI counts assigned to the B6 allele (Eq. 3).

21. Page 28. In Eq. 5 write XP_c and not XP_c , or something simpler, and place 100 on the left of the parenthesis as $100 \times (...)$ or better as $100 \times \max\{0, 1 - \sum ... / \sum ... \}$. Define terms.

We have changed this equation according to the reviewer’s suggestions and defined all terms of the formula in the text following the equation.

22. Page 29. Why using a threshold of 10% and not any other threshold, say 5% or 20%? Before you used XP_c and now you are using XP_b . You should not be doing that, it is confusing and mathematically wrong. Same issue in $X_{b,g}^{Xi}$ and $X_{b,g}^{Xa}$. Define these terms.

We have now repeated the analysis for a series of XP_c threshold values between 0 and 20 to show that the genes identified in this analysis are robust to the parameter choice. This new analysis is presented in Fig. S9c. We apologize for the confusion regarding the terms XP_c and XP_b . We have extended the section on the differential silencing analysis substantially (p. 32-34) and have tried to explain more clearly that XP_c and XP_b refer to two quantities with different meaning: while XP_c is the XCI progress computed for each single cell c (which is a proxy for the extent of inactivation that has already occurred on the inactivated allele), XP_b refers to the overall XCI progress for a group of cells in one bin b , where total gene read counts have been lumped across all cells belonging to that bin in order to obtain a more robust fitting for the silencing dynamics of each gene in equation 11. The terms $X_{g,b}^{Xi}$ and $X_{g,b}^{Xa}$ are not used any more in the manuscript.

23. Pages 29-30: Define terms in Eq. 6. The description after Eq. 6 and to the end of Methods on page 30 is difficult to follow due to complex and difficult to read notation. It must be revised. You really need to try to make the math readable and understandable.

We have rewritten the section extensively (p. 32-34). We have now clarified all terms in Eq.6 (now Eq. 7) and gave also an intuitive explanation of the $r_{g,b}$ quantity as ‘the extent of inactivation of the g -th gene in the b -th bin’ and that ‘Intuitively, a value of $r_{g,b}$ close to zero indicates that the g -th gene has been completely silenced on the X_i allele, while a value proximal to one indicates that the two alleles have similar gene expression levels’. This amount needs to be corrected for expression differences between the two alleles which are due to the different genotypes and not to X-chromosome inactivation. Therefore we divided the ratio $r_{g,b}$ for each gene by the same ratio for that gene computed from the Xist-negative cells, i.e. cells that do not express Xist. We have tried to clarify this now.

We explain how the allele-specific (normalized) inactivation ratio of a X-linked gene was modeled as a function of the total inactivation already occurred on that allele with an exponential decay (or loglinear) function and how we assessed for significant differences in silencing of individual genes. We explain more in detail every term and also provide an intuitive explanation for it.

Reviewer #2 (Remarks to the Author):

This is an interesting and well performed study of X chromosome inactivation in a mouse pluripotent stem cell system with allelic resolution. A number of novel features of XCI are revealed, including the high-resolution temporal dynamics of XCI with single-cell resolution, the identification of putative Xist regulators and differences between genetic background. The manuscript is well written.

I have some questions which upon reading remained open for me:

1. X chromosome upregulation. Can you also refer to the Lentini/Reinus BioRxiv 2020 work and indicate where your findings corroborate or differ from this other study? The second paragraph of page 6 may be a good place to further discuss this.

We have added a discussion of Lentini et al, BioRxiv, 2020, which can be found at the top of p. 21 of the revised version of the manuscript.

2. Why is the upregulation of Xa not shown in Figure 1A?

The Scheme in Fig. 1A summarizes what was previously known about X-chromosome regulation during ES cell differentiation. Since X-upregulation in female cells not undergoing XCI is one of the results of our study, it is not shown in that schematic.

3. Figure 1. Can you show a UMAP and project the X/A ratio on it? One plot for Xist+ cells and one for Xist- cells?

Following the suggestion of the reviewer we now show such a plot in Fig. 1g.

4. Figure 4E: difficult to read because it is not clear if the legend belongs to the figure or not and it takes a while to figure out.

To improve readability of the figure we have modified the symbols in the legend to better reflect those shown in the plots b and e and pointed towards the legend in the caption.

5. Page 7, penultimate line: replace 'reveal' by 'confirms'

We have modified the text as suggested (now on p. 8).

6. Page 11: last paragraph -> We then integrated 'X' chromosome-wide.... Specificity it is the X chromosome.

We have modified the text as suggested (now on p. 9).

7. Page 13, third sentence before last "XCI and differentiation appeared to be partially independent process". It was not clear to me how the figure shows that.

We have deleted that sentence in the revised version of the manuscript.

8. Can you calculate allele-specific X/A ratios (X_{Cast}/A and X_{Mus}/A ratios) and plot that along the pseudotime for cells that inactivate the Mus or the Cast allele (2 plots)? This could go into Figure 5 or 1.

Following this suggestion we are now showing the allele-specific X:A ratio on the X_a and X_i in Supplemental Figure S6a.

9. Profiling random XCI in more physiological contexts, including primary human cells in vivo. How will the experimental and computational strategies developed here help? What if there are much less SNPs in humans and these are not always known? Can you also mention the expected limitations of the human system? You may also want to refer to Tukiainen/MacArthur Nature 2017.

We thank the reviewer for pointing out the missing reference. The reference is now cited in the last paragraph of the discussion section together with a more detailed explanation on how we envision to apply the approaches used in our study to primary human cells (p. 22).

10. Page 24. First paragraph 'genome genome'.

We have corrected this mistake.

11. Page 26. X:A Ratios. Were non-expressed genes removed from the analysis of X:A ratios?

Yes, lowly expressed genes, which were detected in <20% of cells, were excluded from all analyses as described in the section "Gene filtering". For clarity this is now explicitly stated within the section "X:A ratio" in the methods section.

12. Figure S1C. Remove extra dotted line on the right.

Done

Reviewer #3:

Reviewer report for “Integrated analysis of Xist upregulation and gene silencing at the onset of random X-chromosome inactivation at high temporal and allelic resolution” by Pacini et al.

SUMMARY:

In this study Pacini et al aimed to answer questions related to the establishment of random X-chromosome inactivation (XCI), more specifically how Xist expression features (level, allelic transcription, and speed of upregulation) affect silencing and what factors trigger Xist expression.

The authors used mouse embryonic stem cells (mESCs) derived from F1 hybrid embryos (Castaneous x C57/Bl6; providing allelic resolution of gene expression for SNP-containing transcripts) that were differentiated in vitro to induce XCI. Individual cells were captured throughout differentiation (Day: 0, 1, 2, 3, 4) using a C1 Fluidigm and were subjected to single-cell RNA-sequencing (scRNAseq) by a 3' strand-sensitive method (C1-HT). Bioinformatical analyses were used to identify allelic expression patterns that correlated with Xist expression features. Finally, the authors used CRISPR/Cas9-mediated genome editing on mESCs to generate lines with deletions of the X-chromosome inactivation center (Xic), including the Xist gene and other genes in Xic (deletion of ~800k region), at either the CAST or Bl6 allele. This system of predictably skewed XCI was used to perform bulk analysis of selected genes using pyrosequencing.

The manuscript is well written and experiments as well as analyses are intuitive and sound. The generated dataset is valuable for the community studying XCI and other allelic events during mESC differentiation. Overall, I find the study valuable and that the manuscript provides novel findings, and I am therefore positive to seeing it becoming published. There is however room for improvements by reinforcing some of the findings. I also find some conclusions a bit overreached and that interpretations based on observed correlations should be moderated. Below you will find points that I think should be addressed.

MAJOR POINTS:

1. The authors should be more explicit to the reader about the benefits as well as the drawbacks of using a 3' method for the allelic scRNAseq analyses. This distinction becomes particularly important as similar allelic scRNAseq studies of in vitro and in vivo differentiation using the same F1 cross have been performed by others but using full-length-coverage smart-seq2 (full-length and high sensitivity; G. Chen Genome Research 2016, S. Cheng

Cell Reports 2019) or smart-seq3 (full-length, high sensitivity, 5' strand specificity; A. Lentini bioRxiv 2020). In the context of the paper, I find strand-specificity and relatively good Xist detection to be the main benefits of the used 3' C1-HT method, while the drawback being lower sensitivity to detect genes as well as much lower number of genes detected at allelic resolution (due to lack of SNPs in read-covered 3' regions). Importantly, strand specificity allows a direct distinction between Xist and antisense Tsix transcripts, which is not possible for smart-seq2 data (although transcripts can be bioinformatically distinguished based on known exon/intron structures and junctions). Also, Xist is known not to be detected efficiently by smart-seq2, likely due to full-length amplification of the method together with Xist transcript's extreme length as well as possible inaccessibility of Xist RNA for reverse transcription (being a chromatin-bound lncRNA). Judging from the data, Xist seems to be better detected by the 3' C1-HT method. If the authors agree on these points, it would strengthen the paper to better highlight the mentioned strong points of the 3' C1-HT and that this distinguishes the newly generated scRNAseq data from that of previous studies. Equally important is to be transparent with the method's drawback (mainly lower sensitivity and lower number of genes with allelic resolution) to the read. This is easily fixed by rewording a few sentences, but I marked this as a major point because many readers may lack sufficient knowledge in details of scRNAseq methodology to appreciate these issues.

We agree that it is important to draw a clear picture on how the technology used compares to previous studies. We have now stated the difference to the previously used full-length methods regarding strand-specificity in the first paragraph of the results section (p. 3). Moreover, we also discuss the advantages and disadvantages of the different approaches in the second paragraph of the discussion section (p. 19).

2. It would be valuable to get an idea about Xist cloud (or speck) formation especially for cells/stages in which Xist is biallelically transcribed, as such cells were shown to have biallelic reduction of X-gene expression. Showing strand-specific Xist RNA-FISH data at the stage of highest biallelic Xist frequency would strengthen the paper. However, if the authors are convinced that RNA-FISH data shown in previous studies performed by themselves or others sufficiently covers this and that limited value would be gained by new RNA-FISH data, the authors can instead more clearly describe previous findings of biallelic Xist cloud formation in the text.

Following the reviewer's suggestion, we have now included RNA-FISH data for day 2,3 and 4 of differentiation, quantified in 3 biological replicates. The results are shown in Fig. 2c and mirror closely the observations made by scRNA-seq.

3. Nanog as the initial trigger of Xist upregulation: Using two different approaches, the authors propose downregulation of Nanog as the earliest event in XCI. However, I think the authors are over-extrapolating the data here as I do not see how they can pinpoint Nanog as the "initial" trigger of Xist upregulation based on the observed correlations. How about all other potential factors that might have been missed (not detected)? I also think that usage of terminology such as "initial" trigger is both risky and unnecessary, because unknown upstream as well as intermediate factors might exist. Please provide full evidence that

Nanog is truly the initial trigger or moderate your conclusions and statements according to the evidence.

We agree that we only provide correlative evidence pointing towards an early role of Nanog in Xist upregulation. We have removed the notion of the “initial trigger” from the manuscript. We now conclude that “down-regulation of naive pluripotency factors, in particular Nanog and up-regulation of early differentiation factors, such as Pou3f1 and Dnmt3a/b seems to initiate XCI” (results, p.13) and that “... initial Xist upregulation is linked to downregulation of naive pluripotency factors, such as Nanog” (discussion, p. 20).

3b. Additional insights might be provided by RNA-seq using the TXdelta cell lines described in the last section of the study.

We have now performed RNA-sequencing in the TXdeltaXic lines to better estimate the chromosome-wide silencing dynamics. The new data is presented in new Figure 7b-d, substituting for the previously shown pyro-sequencing data (now in Suppl Fig. S11). These results show clearly more rapid silencing of the Cast compared to the B6 chromosome, and validate the detected strain-specific escapees.

MINOR POINTS

1. That XCI is partially initiated (reduced gene expression) on both alleles in cells expressing Xist biallelically is clearly a main finding of the study. This finding should be highlighted in the abstract.

In the revised version of the manuscript we now explicitly describe this observation in the abstract.

2. Implementation of computational methodology to analyze XCI at the single-cell level: The use of RNA velocity to look at allelic silencing is one of the most exciting but is only used for visualization. I think it could strengthen the manuscript if the output from this analysis section was utilized more in the manuscript.

In the revised version of the manuscript we developed an approach, which uses RNA-velocity to identify putative XCI regulators, shown in new Fig. 5+S7 and described on p.10-13. We group cells according to their predicted X-chromosomal expression change from the RNA-velocity estimation (since XCI leads to gene down-regulation). Through differential expression analysis between the groups and correlation analysis we then identify putative regulators of XCI, which show a good overlap with the Xist-based analyses, as shown in new Figure 5.

3. It would be nice to see pseudotime results incorporated with the analysis to show how the allelic expression of X-genes behave during the differentiation process rather than relative to Xist. Differential expression could also be performed along parts of the pseudotime

trajectory using MAST to identify early events in a manner that is less dependent on data timepoints as XCI is variable.

Accounting for differentiation heterogeneity through pseudotime is in general a very useful suggestion. In our specific dataset, however, the pseudotime correlates strongly with real time, but only weakly with XCI initiation, as shown in new Fig. S6b and Fig. 6c. This suggests that differentiation occurs in a fairly homogeneous manner with XCI initiation being highly variable. Unfortunately this heterogeneity does not allow us to perform the proposed analysis. To clarify this we have removed a sentence, suggesting pronounced heterogeneity in the pseudotime analysis and have pointed out XCI heterogeneity explicitly on p. 10.

4. Estimation of silencing dynamics. The authors estimate the silencing rate of individual genes for each allele along the XCI process and identify both gene- and strain-specific effects which is supported by reanalysis of published data as well as their experimental cell line model. Here, it would be interesting to see how the silencing rate behaves across the X chromosome as previous papers have suggested that silencing occurs more rapidly in close vicinity to Xist whereas this may not be true at the single-cell level. Does the 3' data have enough gene coverage with allele-resolved expression to perform this analysis?

Following the reviewer's suggestion, we have now analyzed how gene silencing rates (XP_{50} values) relate to the genomic distance from Xist, shown in new Fig. 6f. In agreement with previous studies, gene silencing tends to occur faster in closer genomic proximity to Xist. Ideally, we would of course like to observe this trend in individual cells, but the number of high-confidence X-linked genes/SNPs per cell is too low for such an analysis.

5. Biallelic Xist expression together with decreased expression of both alleles have been reported in female human preimplantation development as well as in human ESCs, i.e. so called "dampening" which would be in line with partial XCI on both X-alleles [S. Petropoulos Cell 2016; A Sahakyan Cell Stem Cell 2017]. In the context of main results of the current study, i.e. partial XCI and reduced expression from both alleles in female mouse cells with biallelic Xist expression, it would be interesting to read a discussion paragraph on potential translation between the observations in human and mouse.

We are happy to follow this suggestion and have added this aspect to the discussion on p. 20.

6. I appreciate the Methods section describing the computational calculations in detail, but I did not find a link to the computational code. Please make the computational code available.

All code used in our study can be found on the Github repository https://github.com/EddaSchulz/Pacini_paper. A code availability statement has been added to the manuscript on p. 36

7. Fig. 1F: The Xist- trend here is quite interesting. Would be good to know if these cells also downregulate Nanog and if this trend is chromosome-wide or driven by few genes.

To address this question we show the expression of pluripotency (Nanog, Esrrb) and differentiation markers (Dnmt3a) in new Fig. S3c. Dnmt3a is upregulated and Nanog and Esrrb are downregulated both in Xist⁺ and Xist⁻ cells, suggesting that both cell groups differentiate. However, as also expected from the differential expression analysis in Fig. 4 the levels of these markers are significantly different between the two groups at several time points. To investigate whether the upregulation on Xist⁻ cells is a chromosome-wide trend, we calculated the mean expression of each X-chromosomal gene relative to autosomal expression, normalized to day 0. This analysis is presented in new Fig. S3b and shows that the median across genes increases significantly over time in Xist⁻ cells, pointing towards a chromosome-wide trend.

8. p.6 “Around 4% of all reads could be mapped in an allele-specific manner (Fig. S1A).” It should also be stated how many genes per cell were detected at the allelic resolution (i.e. having coverage over SNP positions).

This data is now provided in Fig. S4b showing that a median of around 2000 genes is detected per cell in the allele-specific analysis.

9. p.8. “We used two different approaches to identify such candidate regulators, based on differential expression and correlation analysis, respectively, to ensure robustness of the results (Supplemental Table S3).” These are both indirect approaches. An alternative approach would be to stratify expression over pseudotime to find genes that react prior to Xist. The authors could consider to use MAST for this after excluding day0 cells and genes with the highest negative FC would be possible candidates to check.

Unfortunately, pseudotime does not correlate very strongly with Xist up-regulation and XCI initiation (Fig. S6b + 6c), as also discussed in our response to point 6, making it difficult to perform the suggested analysis. However, as an independent measure of XCI initiation, we have now performed differential expression and correlation analyses based on RNA velocity estimates as shown in Fig. S7. The results are in good agreement with the Xist-based analyses as shown in Fig. 5.

10. p.20 More Xist from B6, can you really say this? Is this not confounded by potential preferential detection (binding, accessibility) between the strains? How did it look in your bulk data?

We want to thank the reviewer for pointing out this potential caveat. We have now analyzed the relative expression from the two alleles in bulk RNA-seq data generated in parallel to the scRNA-seq experiment and could indeed not confirm significantly higher Xist levels from the B6 compared to the Cast chromosome (new Fig. S5b). Although for the deltaXic lines there seems to be a (non-significant) trend towards higher Xist expression from the B6 allele at day 1 (Fig. 7b, Fig. S11b), we removed this conclusion from the manuscript and discuss this discrepancy on p. 8.

FINAL REMARKS: I hope these comments will help the authors to further improve the manuscript. I find the study and data very valuable.

REVIEWER COMMENTS

Reviewer #1 (Remarks to the Author):

I would like to congratulate and thank the authors for the effort they put for improving their paper. The new version addresses my previous comments, and I have no problem recommending publication. Below, I list some points that must be taken care of.

1. Page 6, write: ($p < 0.001$, Mann Whitney U two-sided test).
2. Page 12. The scheme you use to display the results in Figs. 4(e,f) and S7 is different than the one you use in Figs. 5 and S8 (white dots vs. white stars, small colored squares vs. large colored rectangles). I think you should unify the schemes used. I personally like the one you used in Figs. 5 and S8, since it is easier to see (although I will suggest that you use dots and not stars to be easier to identify).
3. Page 12, caption in Fig. 5, write: "... and Spearman's correlation coefficient ρ , in (c), are shown." Using (ρ, c) is confusing mathematically. Similarly, write "... the BH-corrected p-value (FDR), in (a), the ...", as well as write "... fold change \log_2FC , in (b), ..." This issue may also be present in other captions, so check carefully.
4. Page 15, replace the two instances of p values with $p < 0.001$.
5. Is the BH-corrected p-value the same as the FDR? The way you write it in several places in the text, you imply that they are the same. However, I think that the BH-corrected p-value provides an upper bound of the FDR and may not be the same as the FDR. In my opinion, you must be careful how you report the BH-corrected p-values through the manuscript (sometimes you say BH-corrected p-value ≤ 0.05 , sometimes you say BH-corrected p-value (FDR) < 0.05 , and sometimes you say FDR = 0.05 or FDR ≤ 0.05). I think this is very confusing. A way to resolve this issue is to refer to "BH-corrected p-value (q-value)" the first time and then refer to q-values subsequently.
6. I still have issues with the text regarding computational methods. Things are still quite confusing due to chosen notation. For this reason, I appended my remarks below hoping they will help.

Reviewer #2 (Remarks to the Author):

I am satisfied with the revised version of the manuscript.

Minor comment.

page 22, second paragraph, add a dot at the end of the sentence on line 4.

Reviewer #3 (Remarks to the Author):

I was glad to see that the revised version of Pacini et al addresses nearly all reviewer points with new data, analyses and rephrasing of text. This paper will be valuable for the XCI community and I hope to see it published soon. I have two (minor) remaining points that I think should be addressed in the final version.

1.
Page 1: "Female mammals carry two X chromosomes..."
This is incorrect or imprecise, as not all mammals have XX / XY sex-chromosome setup. I suggest changing "mammals" to "therian mammals" to be more correct on this part.
- 2.

Discussion, Page 21:

Paragraph 1:

“Xist upregulation was followed by progressive gene silencing of X-linked genes, resulting in a decrease of the X-to-autosome ratio. At the same time the X:A ratio increased in cells that did not upregulate Xist, likely reflecting the process of X upregulation [30–35].”

Paragraph 2:

“A recent study, which also analyzed X upregulation in differentiating female mESCs through scRNA-seq reported X upregulation only on the active X of cells that underwent XCI (XaXi), but not in cells that failed to initiate XCI (XaXa). Since that study classified cells into XaXa and XaXi using the X-chromosomal allelic ratio, and not Xist expression patterns, the XaXa-classified cells might not necessarily correspond to Xist-negative cells, but might in part have initiated XCI on both alleles (XiXi), which could explain, why upregulation of the X-chromosome was not observed.”

I find the second paragraph unnecessarily speculative and very risky without providing any re-analysis of the Lentini data. The referred Lentini study is an early-stage manuscript (biorxiv preprint) and an updated or published version may or may not turn out to demonstrate upregulation or the lack thereof in XaXa cells classified according to allelic Xist patterns. I strongly suggest removing this speculative paragraph, particularly since it does not add any substantial information or insights to the reader. As there is an overall agreement on most overlapping points in Pacini and Lentini, the citation could be moved to the previous paragraph alternatively removed.

Reviewer #1

I would like to congratulate and thank the authors for the effort they put for improving their paper. The new version addresses my previous comments, and I have no problem recommending publication. Below, I list some points that must be taken care of.

1. Page 6, write: ($p < 0.001$, Mann Whitney U two-sided test).

We have substituted " $p < 6.2 \cdot 10^{-14}$ " with " $p < 0.001$ ", as suggested by the reviewer.

2. Page 12. The scheme you use to display the results in Figs. 4(e,f) and S7 is different than the one you use in Figs. 5 and S8 (white dots vs. white stars, small colored squares vs. large colored rectangles). I think you should unify the schemes used. I personally like the one you used in Figs. 5 and S8, since it is easier to see (although I will suggest that you use dots and not stars to be easier to identify).

In Figure 5 we chose a different representation of the data compared to Fig. 4 and S7, because in the plot type used in Fig. 4 and S7, where the value is also encoded in the object size, the dot, which represents statistical significance, is only visible, if the object has a certain size. We cannot really follow the reviewer's argument here, why we should not use different plots in the different figures.

3. Page 12, caption in Fig. 5, write: "... and Spearman's correlation coefficient ρ , in (c), are shown." Using (ρ, c) is confusing mathematically. Similarly, write "... the BH-corrected p-value (FDR), in (a), the ...", as well as write "... fold change \log_2FC , in (b), ..." This issue may also be present in other captions, so check carefully.

We have modified the figure legends accordingly.

4. Page 15, replace the two instances of p values with $p < 0.001$.

Done.

5. Is the BH-corrected p-value the same as the FDR? The way you write it in several places in the text, you imply that they are the same. However, I think that the BH-corrected p-value provides an upper bound of the FDR and may not be the same as the FDR. In my opinion, you must be careful how you report the BH-corrected p-values through the manuscript (sometimes you say BH-corrected p-value ≤ 0.05 , sometimes you say BH-corrected p-value (FDR) < 0.05 , and sometimes you say FDR = 0.05 or FDR ≤ 0.05). I think this is very confusing. A way to resolve this issue is to refer to "BH-corrected p-value (q-value)" the first time and then refer to q-values subsequently.

We have replaced "FDR" throughout the manuscript with "BH-corrected p-values".

6. I still have issues with the text regarding computational methods. Things are still quite confusing due to chosen notation. For this reason, I appended my remarks below hoping they will help.

We have modified the methods section, following most suggestions of the reviewer.

Reviewer #2

I am satisfied with the revised version of the manuscript.

Minor comment.

page 22, second paragraph, add a dot at the end of the sentence on line 4.

We have modified the text as suggested by the reviewer.

Reviewer #3

I was glad to see that the revised version of Pacini et al addresses nearly all reviewer points with new data, analyses and rephrasing of text. This paper will be valuable for the XCI community and I hope to see it published soon. I have two (minor) remaining points that I think should be addressed in the final version.

1.

Page 1: "Female mammals carry two X chromosomes..."

This is incorrect or imprecise, as not all mammals have XX / XY sex-chromosome setup. I suggest changing "mammals" to "therian mammals" to be more correct on this part.

We have substituted "mammals" with "therian mammals" as suggested by the reviewer.

2.

Discussion, Page 21:

Paragraph 1:

"Xist upregulation was followed by progressive gene silencing of X-linked genes, resulting in a decrease of the X-to-autosome ratio. At the same time the X:A ratio increased in cells that did not upregulate Xist, likely reflecting the process of X upregulation [30–35]."

Paragraph 2:

"A recent study, which also analyzed X upregulation in differentiating female mESCs through scRNA-seq reported X upregulation only on the active X of cells that underwent XCI (XaXi), but not in cells that failed to initiate XCI (XaXa). Since that study classified cells into XaXa and XaXi using the X-chromosomal allelic ratio, and not Xist expression patterns, the XaXa-classified cells might not necessarily correspond to Xist-negative cells, but might in part have initiated XCI on both alleles (XiXi), which could explain, why upregulation of the X-chromosome was not observed."

I find the second paragraph unnecessarily speculative and very risky without providing any re-analysis of the Lentini data. The referred Lentini study is an early-stage manuscript (biorxiv preprint) and an updated or published version may or may not turn out to demonstrate upregulation or the lack thereof in XaXa cells classified according to allelic Xist patterns. I strongly suggest removing this speculative paragraph, particularly since it does not add any substantial information or insights to the reader. As there is an overall agreement on most overlapping points in Pacini and Lentini, the citation could be moved to the previous paragraph alternatively removed.

We have removed the paragraph from the revised version of the manuscript.